# MUC1-C regulates lineage plasticity driving progression to neuroendocrine prostate cancer

Yota Yasumizu[1], Hasan Rajabi[1], Caining Jin[1], Tsuyoshi Hata[1,6], Sean Pitroda[2], Mark D. Long [3], Masayuki Hagiwara[1], Wei Li[1], Qiang Hu [3], Song Liu[3], Nami Yamashita[1], Atsushi Fushimi[1], Ling Kui[1], Mehmet Samur[1], Masaaki Yamamoto[1,6], Yan Zhang[1], Ning Zhang[1], Deli Hong[1], Takahiro Maeda[4], Takeo Kosaka [4], Kwok K. Wong [5], Mototsugu Oya[4] & Donald Kufe [1]*

Neuroendocrine prostate cancer (NEPC) is an aggressive malignancy with no effective targeted therapies. The oncogenic MUC1-C protein is overexpressed in castration-resistant prostate cancer (CRPC) and NEPC, but its specific role is unknown. Here, we demonstrate that upregulation of MUC1-C in androgen-dependent PC cells suppresses androgen receptor (AR) axis signaling and induces the neural BRN2 transcription factor. MUC1-C activates a MYC→BRN2 pathway in association with induction of MYCN, EZH2 and NE differentiation markers (ASCL1, AURKA and SYP) linked to NEPC progression. Moreover, MUC1-C suppresses the p53 pathway, induces the Yamanaka pluripotency factors (OCT4, SOX2, KLF4 and MYC) and drives stemness. Targeting MUC1-C decreases PC self-renewal capacity and tumorigenicity, suggesting a potential therapeutic approach for CRPC and NEPC. In PC tissues, MUC1 expression associates with suppression of AR signaling and increases in BRN2 expression and NEPC score. These results highlight MUC1-C as a master effector of lineage plasticity driving progression to NEPC.

[1] Dana-Farber Cancer Institute Harvard Medical School, Boston, MA, USA. [2] Department of Radiation and Cellular Oncology, University of Chicago, Chicago, IL, USA. [3] Department of Biostatistics and Bioinformatics Roswell Park Comprehensive Cancer Center, Buffalo, NY, USA. [4] Department of Urology, Keio University School of Medicine Shinjuku-ku, Tokyo, Japan. [5] Laura and Isaac Perlmutter Cancer Center, New York University Langone Medical Center, New York, NY, USA. [6] Present address: Department of Gastrointestinal Surgery, Graduate School of Medicine, Osaka University, Suita, Osaka, Japan. *email: donald_kufe@dfci.harvard.edu

Castration-resistant prostate cancer (CRPC) often progresses to a more aggressive form with neuroendocrine features (CRPC-NE) in association with resistance to androgen receptor (AR) pathway targeted therapy[1–4]. Hallmarks of neuroendocrine prostate cancer (NEPC) include (i) loss of AR axis, p53 and RB signaling, (ii) activation of the neural BRN2 transcription factor (TF) and (iii) increased stemness associated with induction of the epithelial-mesenchymal transition (EMT) and SOX2 expression[5–7]. NEPC has also been linked to the upregulation of MYCN and the Polycomb Repressive Complex 2 (PRC2) component EZH2 (refs. [8–12]). The incidence of NEPC is increasing with the widespread use of AR-targeted agents, such as enzalutamide (ENZ), for CRPC treatment[3,4]. Patients diagnosed with NEPC have a median overall survival of <1 year[3,4]. In this regard, there are presently no effective targeted agents for the treatment of this disease, emphasizing the need for identifying druggable effectors that drive lineage plasticity to NEPC development.

Mucin 1 (MUC1) is a heterodimeric protein that is aberrantly overexpressed in diverse human carcinomas and contributes to hallmarks of the cancer cell, including EMT, stemness, anti-cancer drug resistance, epigenetic reprogramming, and immune evasion[13–16]. The upregulation of MUC1 as found in approximately 90% of PCs is associated with Gleason grades ≥7, aggressive disease and increased risk of recurrence[17–19]. In addition, MUC1 expression has been linked to (i) early biochemical failure and PC-related death[20], and (ii) bone metastases in CRPC[21]. These findings have supported the potential importance of MUC1 in advanced PC; however, there is no known link between MUC1 and PC progression.

MUC1 consists of two subunits; that is (i) an N-terminal highly glycosylated subunit (MUC1-N), which is shed from the cell membrane, and (ii) an oncogenic C-terminal transmembrane subunit (MUC1-C)[13–16]. MUC1-C consists of a 58 aa extracellular domain, a 28 aa transmembrane region and an intrinsically disordered 72 aa cytoplasmic tail[14]. In cancer cells, MUC1-C associates with receptor tyrosine kinases, such as EGFR among others, at the cell membrane and contributes to activation of their downstream pathways[14,16]. MUC1-C is imported into the nucleus, where it interacts directly with TFs, including MYC and p53, and thereby regulates their transactivation functions[16,22,23]. MUC1-C is also involved in epigenetic reprogramming by activating (i) DNA methyltransferases (DNMTs), (ii) components of PRC1/2, including EZH2, and (iii) the NuRD chromatin remodeling complex, further supporting a role for MUC1-C in gene regulation, including the repression of tumor suppressor genes (TSGs)[15,16,23,24].

There is no known association for MUC1-C with BRN2, neuroendocrine differentiation or lineage plasticity in PC. The present studies demonstrate that MUC1-C suppresses AR axis signaling in PC cells and drives expression of the BRN2 neural TF by a previously unreported MYC-dependent mechanism. We also show that MUC1-C (i) activates the BRN2 pathway in association with induction of MYCN, EZH2, and NE markers linked to NEPC progression, (ii) suppresses the p53 pathway, (iii) induces the OCT4, SOX2, KLF4 and MYC (OSKM) pluripotency factors and (iv) drives stemness. In support of clinical relevance, we report that targeting MUC1-C in vitro and in PC tumor xenograft models inhibits BRN2 signaling, the NE phenotype, self-renewal capacity and tumorigenicity.

## Results

**MUC1-C expression is linked to to androgen independence and self-renewal.** C4-2B prostate cancer cells were previously generated from androgen-dependent LNCaP cells selected in vivo under conditions of androgen ablation[25]. Here, C4-2B cells were selected for long-term culture in phenol red-free medium and charcoal-stripped serum to assess the potential for MUC1-C involvement in supporting androgen-independent (AI) growth. In contrast to LNCaP and C4-2B cells, the selected androgen-independent LNCaP cells (designated LNCaP-AI) proliferate under these androgen-depleted conditions (Fig. 1a). AR expression was decreased in LNCaP-AI, as compared to C4-2B and LNCaP, cells (Fig. 1b). AR axis signaling was also downregulated in LNCaP-AI cells as evidenced by (i) decreases in PSA/KLK3 and NKX3.1 mRNA (Supplementary Fig. 1a, b) and protein (Fig. 1b), and (ii) resistance to treatment with the antiandrogen enzalutamide (ENZ) (Supplementary Fig. 2), which distinguish CRPC with NE features (CRPC-NE) from prostatic adenocarcinoma[26]. As examined by phase contrast microscopy, the LNCaP-AI cells exhibit distinct patterns of growth with the formation of clusters compared to that found for C4-2B cells (Supplementary Fig. 3a). Staining with H&E further demonstrated that C4-2B cells have dense round or oval nuclei with diffuse chromatin and the absence of distinct nucleoli (Supplementary Fig. 3b, left panels). In contrast, the LNCaP-AI cells were found to have larger irregular nuclei, visible nucleoli and occasional giant cells with smudgy chromatin, similar in part with morphologic features identified in certain small cell carcinomas of the prostate[27] (Supplementary Fig. 3b, right panels). We also found that LNCaP and C4-2B cells have low levels of MUC1-C expression and that MUC1-C is significantly upregulated in LNCaP-AI cells (Fig. 1c, left and right). To investigate the functional significance of these observations, we established LNCaP-AI cells expressing a tet-inducible control shRNA (LNCaP-AI/tet-CshRNA) or a MUC1-CshRNA (LNCaP-AI/tet-MUC1shRNA). Treatment with doxycycline (DOX) resulted in downregulation of MUC1-C in LNCaP-AI/tet-MUC1shRNA, but not LNCaP-AI/tet-CshRNA, cells (Fig. 1d). DOX treatment of LNCaP-AI/tet-MUC1shRNA cells was also associated with inhibition of growth (Fig. 1e), invasion (Fig. 1f), colony formation (Fig. 1g) and tumorsphere formation (Fig. 1h), supporting the notion that MUC1-C is of importance for the malignant phenotype of these cells.

**MUC1-C induces BRN2 and NE differentiation.** To search for further evidence linking MUC1-C with the AI phenotype, RNA-seq was performed on control and DOX-treated LNCaP-AI/tet-MUC1shRNA cells. Analysis of the data using the MSigDB Hallmark Gene Set showed that MUC1-C plays a significant role in suppression of the AR response (Fig. 2a) and that silencing MUC1-C is associated with induction of PSA/KLK3, NKX3.1 and TMPRSS2 expression (Fig. 2b). Suppression of AR signaling in LNCaP-AI cells was associated with upregulation of (i) BRN2, a neural TF and driver of the NE phenotype[7] (Fig. 2c, d), and (ii) MYCN and EZH2 (Fig. 2d), which have been linked with progression to CRPC with neuroendocrine features (CRPC-NE)[8–12]. Silencing MUC1-C in LNCaP-AI cells resulted in the downregulation of BRN2 mRNA levels (Fig. 2e) and decreases in BRN2, MYCN and EZH2 protein (Fig. 2f). Silencing MUC1-C also suppressed achaete-scute homolog 1 (ASCL1), aurora kinase A (AURKA) and synaptophysin (SYP) expression (Fig. 2g), which have been linked to progression of CRPC to NEPC[8].

**MUC1-C induces *BRN2* by a MYC-mediated mechanism.** *BRN2* is repressed by an AR-mediated mechanism in PC cells[7]. Accordingly, one explanation for the observation that MUC1-C drives BRN2 expression is that MUC1-C suppresses AR and in turn AR-mediated repression of the *BRN2* gene. Indeed, AR occupancy on the *BRN2* promoter was decreased in LNCaP-AI, as compared to LNCaP, cells (Fig. 3a). Additionally, while

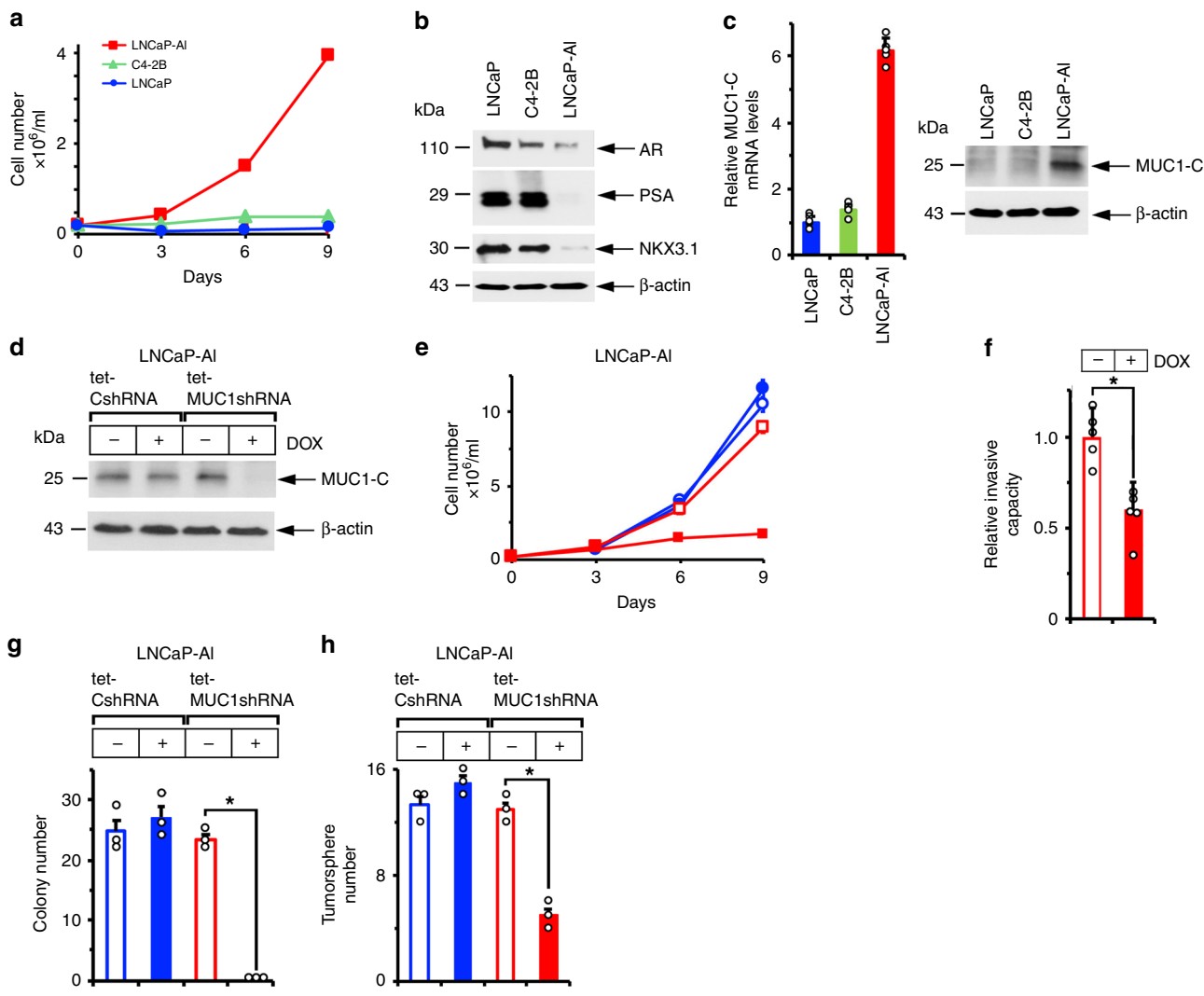

**Fig. 1 MUC1-C drives AI and self-renewal capacity. a** LNCaP (blue circles), C4-2B (green triangles) and LNCaP-AI (red squares) cells were cultured in androgen-depleted medium for 10 days, seeded at $2 \times 10^4$ cells/ml and then monitored for cell growth. Cell number (mean of three biologic replicates) was determined by trypan blue staining. **b** Lysates from LNCaP, C4-2B and LNCaP-AI cells were immunoblotted with antibodies against the indicated proteins. **c** LNCaP, C4-2B and LNCaP-AI cells were analyzed for MUC1-C mRNA levels by qRT-PCR using primers listed in Supplementary Table 1. The results (mean±SD of four determinations) are expressed as relative mRNA levels compared to those obtained for LNCaP cells (assigned a value of 1)(left). Lysates were immunoblotted with antibodies against the indicated proteins (right). **d** LNCaP-AI cells stably expressing a tet-CshRNA or tet-MUC1shRNA were treated with vehicle or 500 ng/ml DOX for 7 days. Lysates were immunoblotted with antibodies against the indicated proteins. **e** LNCaP-AI/tet-CshRNA (blue circles) and LNCaP-AI/tet-MUC1shRNA (red squares) cells seeded at $2 \times 10^4$ cells/ml in androgen-depleted medium were treated with vehicle (open symbols) or 500 ng/ml DOX (closed symbols) for the indicated times. Cell number (mean±SD of three replicates) was determined by trypan blue staining. **f** LNCaP-AI/tet-MUC1shRNA cells treated with vehicle or 500 ng/ml DOX for 7 days were assayed for invasive capacity in matrigel coated transwell chambers. Results (mean ± SD of five determinations) are expressed as the relative invasive capacity compared to that obtained with the control cells (assigned a value of 1). **g** LNCaP-AI/tet-CshRNA and LNCaP-AI/tet-MUC1shRNA cells seeded at 500 cells/well in six-well plates were treated with vehicle or 500 ng/ml DOX. Colonies were stained with crystal violet on day 14. The results are expressed as the colony number (mean±SD of three determinations) per well. **h** LNCaP-AI/tet-CshRNA and LNCaP-AI/tet-MUC1shRNA cells seeded at $5 \times 10^3$ cells/well in ultra-low attachment six-well plates were treated with vehicle or 500 ng/ml DOX for 14 days. The results are expressed as the tumorsphere number (mean±SD of three determinations) per well. *$p < 0.05$ (unpaired Mann–Whitney $U$ test). Dot plots are represented by open circles in the bar graphs. Source data are provided as a Source Data file.

performing these experiments, we detected MUC1-C occupancy on the *BRN2* promoter, invoking the possibility that MUC1-C directly activates BRN2 expression. MUC1-C activates *MYC* expression in certain cancer cells[28–30]. In addition, the MUC1-C cytoplasmic domain binds directly to the MYC HLH-LZ region and, as a result, MUC1-C forms a complex with MYC on the promoters of MYC target genes[23]. Along those lines, we identified putative MYC binding sites in the *BRN2* promoter (Fig. 3b). ChIP studies performed on chromatin from LNCaP-AI cells

demonstrated that MUC1-C and MYC occupy (Fig. 3c, left) and, as evidenced by re-ChIP analysis, form a complex on this region of the *BRN2* promoter (Fig. 3c, right). We also found that silencing MUC1-C decreases MYC occupancy (Fig. 3d), consistent with involvement of MUC1-C in enhancing MYC trans-activation complexes[23]. Functional studies performed with a BRN2 promoter-luciferase reporter (pBRN2-Luc) demonstrated suppression of activity by (i) mutating the distal, but not proximal, MYC binding site (Fig. 3e), and (ii) silencing MUC1-C or

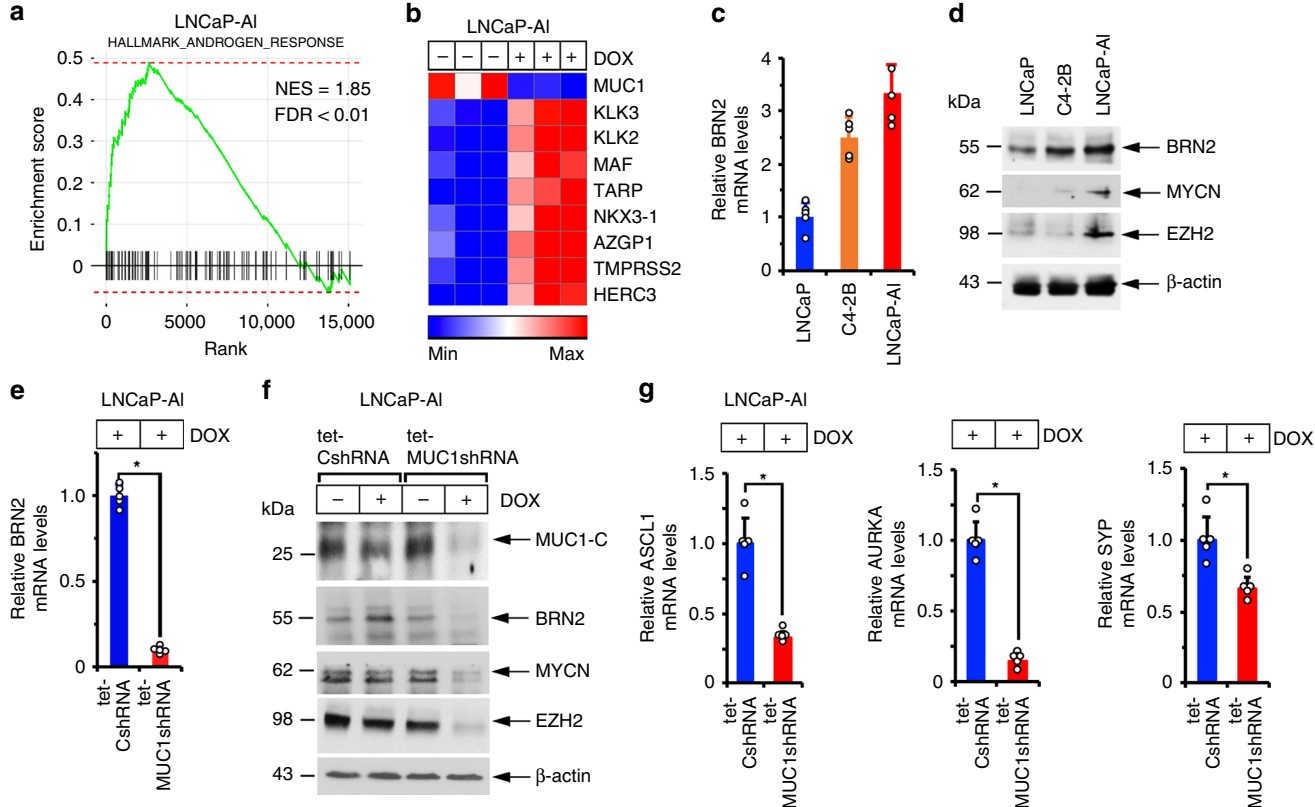

**Fig. 2 MUC1-C induces expression of BRN2 and NE markers. a,b** RNA-seq was performed in triplicate on LNCaP-AI/tet-MUC1shRNA cells treated with vehicle or 500 ng/ml DOX for 7 days. **a** The datasets were analyzed with GSEA, using the Hallmark gene signature collection. Silencing MUC1 was significantly associated with upregulation of the Androgen Response pathway. **b** Heatmap depicting the effects of silencing MUC1 on AR pathway genes. **c** LNCaP, C4-2B and LNCaP-AI cells were analyzed for BRN2 mRNA levels by qRT-PCR. The results (mean±SD of four determinations) are expressed as relative mRNA levels compared to that obtained for LNCaP cells (assigned a value of 1). **d** Lysates from LNCaP, C4-2B, and LNCaP-AI cells were immunoblotted with antibodies against the indicated proteins. **e** LNCaP-AI cells stably expressing a tet-CshRNA or tet-MUC1shRNA were treated with 500 ng/ml DOX for 7 days. BRN2 mRNA levels were analyzed by qRT-PCR. The results (mean±SD of four determinations) are expressed as relative mRNA levels compared to that obtained for DOX-treated LNCaP-AI/tet-CshRNA cells (assigned a value of 1). **f** LNCaP-AI/tet-CshRNA and LNCaP-AI/tet-MUC1shRNA cells were treated with vehicle or 500 ng/ml DOX for 7 days. Lysates were immunoblotted with antibodies against the indicated proteins. **g** LNCaP-AI cells expressing a tet-CshRNA or tet-MUC1shRNA were treated with 500 ng/ml DOX for 7 days. ASCL1 (left), AURKA (middle) and SYP (right) mRNA levels were analyzed by qRT-PCR. The results (mean±SD of five determinations) are expressed as relative mRNA levels compared to that obtained for DOX-treated LNCaP/tet-CshRNA cells (assigned a value of 1). *$p < 0.05$ (Student's $t$-test). Source data are provided as a Source Data file.

MYC (Fig. 3f). Silencing MYC in DOX-treated LNCaP-AI/tet-MYC shRNA cells also decreased BRN2 expression (Fig. 3g), further supporting a model in which MUC1-C drives BRN2 by a MYC-mediated mechanism. By extension, targeting MYC in LNCaP-AI cells with the BET bromodomain inhibitor JQ1 also decreased BRN2 mRNA and protein levels (Supplementary Fig. 4a, b).

**Silencing MUC1-C downregulates BRN2 and self-renewal.** MUC1-C, and not AR, is constitutively expressed in the AI DU-145 cells[31] and NEPC tumor-derived NCI-H660 cells[32] (Fig. 4a), consistent with an inverse relationship between MUC1 and AR in PC cell lines[33] (Supplementary Fig. 5). Accordingly, DU-145 cells expressing tet-CshRNA or tet-MUC1shRNA were studied for effects of MUC1-C silencing on gene expression patterns. As found in the LNCaP-AI cell studies, (i) analysis of the DU-145 RNA-seq data showed that silencing MUC1-C is associated with upregulation of the Hallmark Androgen Response pathway (Supplementary Fig. 6a, b), and (ii) silencing MUC1-C or MYC in DU-145 cells resulted in downregulation of BRN2 (Fig. 4b, c; Supplementary Fig. 5c). We also found that (i) MUC1-C and MYC form a complex on the *BRN2* promoter (Fig. 4d, left and right) and (ii) silencing MUC1-C decreases MYC occupancy

(Fig. 4e). Moreover, silencing MUC1-C in DU-145 cells resulted in inhibition of growth (Fig. 4f), invasion (Fig. 4g) and colony formation (Fig. 4h), consistent with dependence on MUC1-C for driving the NE phenotype and self-renewal.

**MUC1-C drives pathways associated with lineage plasticity.** Lineage plasticity in PC has been linked to suppression of the p53 and RB pathways and to induction of SOX2 expression[5–7,34]. Analysis of our RNA-seq data showed that silencing MUC1-C in LNCaP-AI and DU-145 cells is highly correlated with upregulation of the Hallmark p53 pathway gene set (Fig. 5a, left and right). MUC1-C drives the phosphorylation and inactivation of RB[28,29,35,36], the main binding partner of E2F and key regulator of E2F activity[37]. Alterations in *RB1* are found at high frequencies in advanced PCs and are associated with poor survival[4]. Additionally, E2F is elevated in NEPC and activates target genes, such as *PEG10*, linked to NEPC progression[38]. Here, silencing MUC1-C in LNCaP-AI and DU-145 cells was significantly associated with downregulation of E2F target genes (Fig. 5b, left and right), indicating that, in addition to p53, MUC1-C contributes to regulation of the RB-E2F axis. BRN2 is necessary for SOX2 activation in neural development[39] and in CRPC cells[7]. In a gain-of-function model, we found that overexpression of MUC1-C in

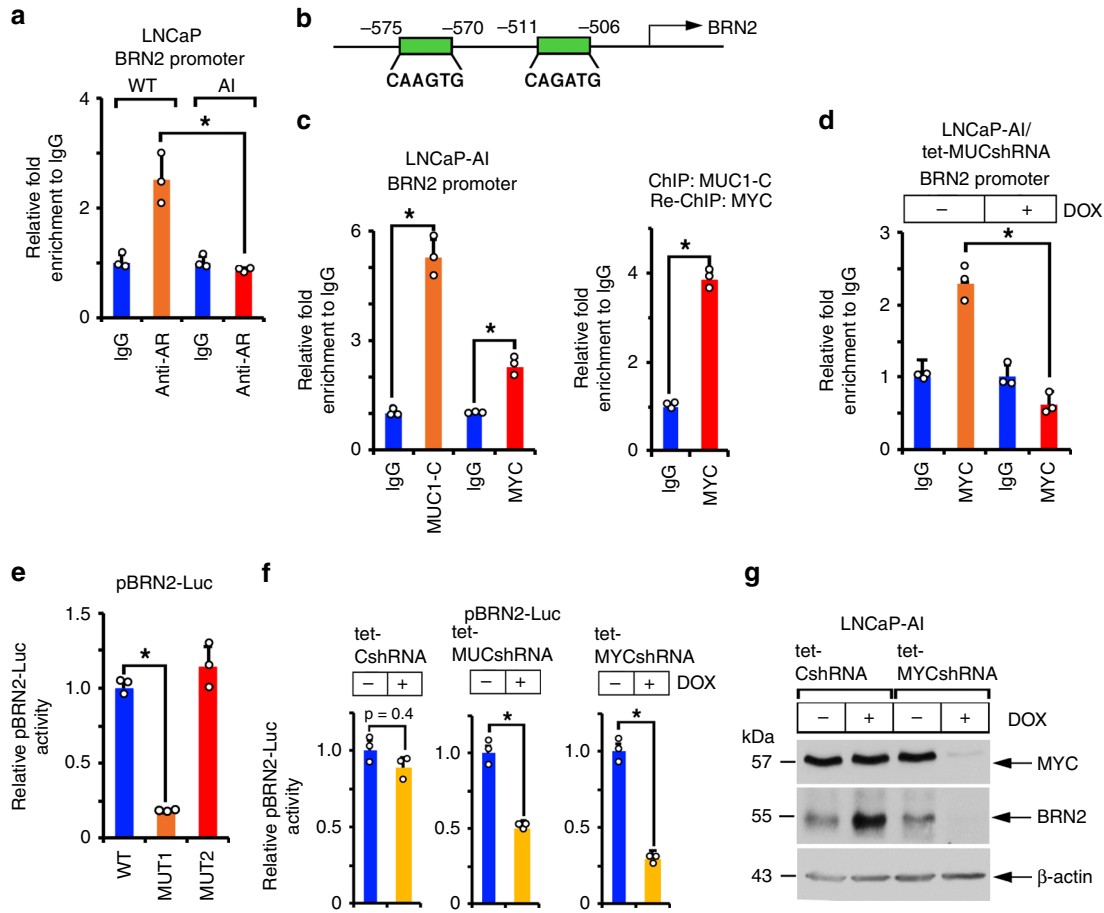

**Fig. 3 MUC1-C induces BRN2 by a MYC-dependent mechanism. a** Soluble chromatin from wild-type (WT) LNCaP and LNCaP-AI cells was precipitated with anti-AR or a control IgG. The DNA samples were amplified by qPCR with primers for the *BRN2* promoter. The results (mean±SD of three determinations) are expressed as the relative-fold enrichment compared to that obtained with the IgG control (assigned a value of 1). **b** Schema of the *BRN2* promoter region with positioning of the putative MYC binding motifs. **c** Soluble chromatin from LNCaP-AI cells was precipitated with anti-MUC1-C, anti-MYC or a control IgG (left). Soluble chromatin from LNCaP-AI cells was precipitated with anti-MUC1-C (ChIP) and then reprecipitated with anti-MYC or a control IgG (re-ChIP) (right). **d** LNCaP-AI/tet-MUC1shRNA cells were treated with vehicle or 500 ng/ml DOX for 7 days. Soluble chromatin was precipitated with anti-MYC or a control IgG. The DNA samples were amplified by qPCR with primers for the *BRN2* promoter. The results (mean ± SD of three determinations) are expressed as the relative-fold enrichment compared to that obtained with the IgG control (assigned a value of 1). **e** LNCaP-AI cells were transfected with pGL3-Basic Luc, pBRN2-Luc (WT), pBRN2-Luc MUT1 or pBRN2-Luc MUT2 for 48 h and then analyzed for luciferase activity. The results (mean±SD of three determinations) are expressed as relative luciferase activity as compared to that obtained for cells transfected with the pBRN2-Luc (WT) vector (assigned a value of 1). **f** LNCaP-AI/tet-CshRNA (left panel), LNCaP-AI/tet-MUC1shRNA (middle panel) and LNCaP-AI/tet-MYCshRNA (right panel) cells treated with vehicle or 500 ng/ml DOX for 5 days were transfected with pGL3-Basic Luc or pBRN2-Luc vectors for 48 h and then analyzed for luciferase activity. The results (mean±SD of three determinations) are expressed as relative luciferase activity as compared to that obtained for untreated cells (assigned a value of 1). **g** LNCaP-AI cells expressing a tet-CshRNA or tet-MYCshRNA were treated with vehicle or 500 ng/ml DOX for 5 days. Lysates were immunoblotted with antibodies against the indicated proteins. *$p < 0.05$ (Student's *t*-test). Source data are provided as a Source Data file.

LNCaP cells increases BRN2 and SOX2 expression at the mRNA (Fig. 5c) and protein (Fig. 5d) levels. Additionally, and like BRN2, we found that silencing MUC1-C in LNCaP-AI cells results in the downregulation of SOX2 expression (Fig. 5e, f), supporting a MUC1-C→MYC→BRN2→SOX2 pathway. SOX2, MYC, KLF4, and OCT4 collectively dedifferentiate fibroblasts to induced pluripotent stem cells (iPSCs) in a manner that is potentiated by p53 and RB suppression[40]. Having demonstrated that MUC1-C induces SOX2 and regulates MYC[28,29], we found that silencing MUC1-C in LNCaP-AI and DU-145 cells decreases expression of the four OSKM pluripotency factors (Fig. 5f, g). Previous findings from BRN2 knockdown and rescue experiments demonstrated BRN2 induction of SOX2 and NE marker expression[7]. In concert with those results, we silenced BRN2 and found downregulation of the NE-associated ASCL1 marker[8] (Fig. 5h). By contrast,

silencing BRN2 had no apparent effect on MYCN (Fig. 5h), which as shown above is driven by MUC1-C signaling. In addition, silencing BRN2 was associated with suppression of SOX2, but not MYC, KLF4 or OCT4 expression (Fig. 5h), indicating that, like MYCN and EZH2, MUC1-C also drives these pluripotency factors by BRN2-independent mechanisms.

**Targeting MUC1-C inhibits PC tumorigenicity.** To extend these experiments to in vivo models, mice bearing established LNCaP-AI/tet-MUC1shRNA tumors were fed DOX to assess effects of MUC1-C on growth and gene expression patterns. DOX treatment was associated with marked inhibition of tumor growth (Fig. 6a). In addition, and consistent with the in vitro results, we found downregulation of (i) MUC1-C, BRN2, MYCN and EZH2,

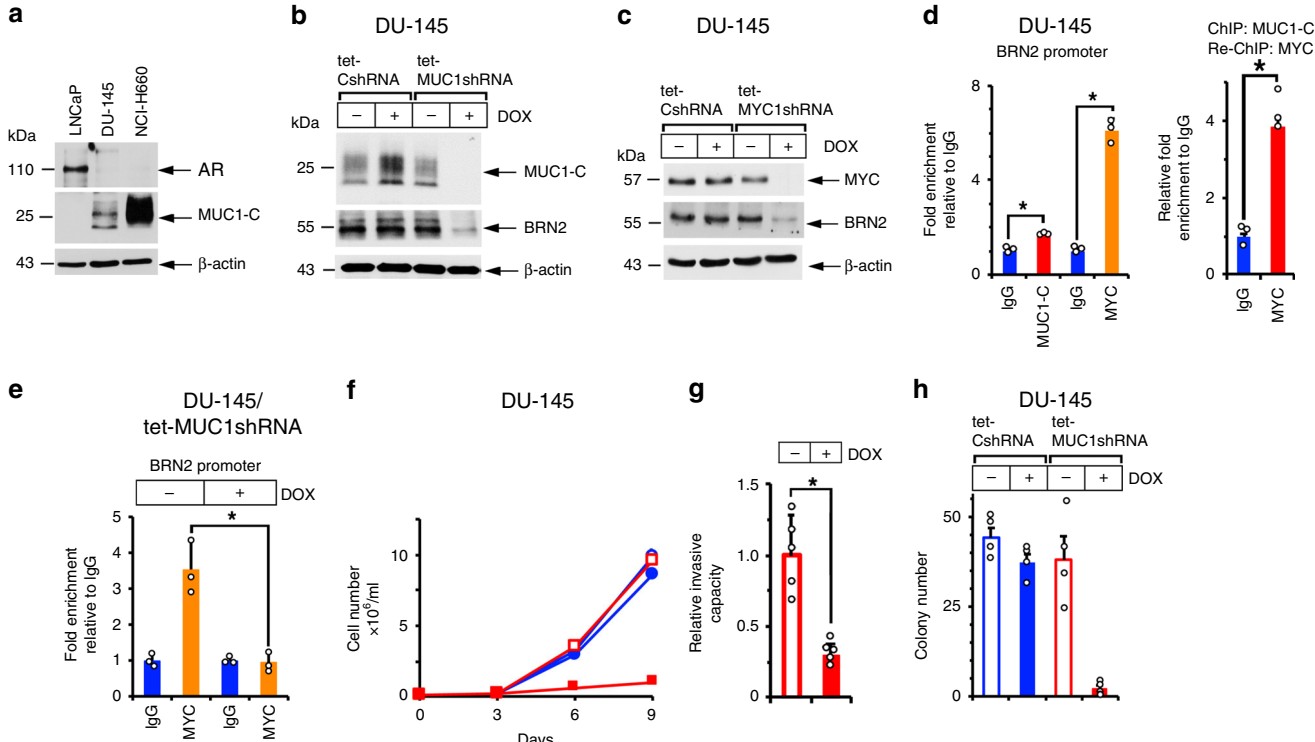

**Fig. 4 MUC1-C drives BRN2, invasion and self-renewal. a** Lysates from LNCaP, DU-145 and NCI-H660 cells were immunoblotted with antibodies against the indicated proteins. **b** and **c** DU-145 cells stably expressing a tet-CshRNA, tet-MUC1shRNA (**b**) or tet-MYCshRNA (**c**) were treated with vehicle or 500 ng/ml DOX for 7 days. Lysates were immunoblotted with antibodies against the indicated proteins. **d** Soluble chromatin from DU-145 cells was precipitated with anti-MUC1-C, anti-MYC or a control IgG (left). Soluble chromatin from DU-145 cells was precipitated with anti-MUC1-C (ChIP) and then reprecipitated with anti-MYC or a control IgG (re-ChIP) (right). **e** DU-145/tet-MUC1shRNA cells were treated with vehicle or 500 ng/ml DOX for 7 days. Soluble chromatin was precipitated with anti-MYC or a control IgG. The DNA samples were amplified by qPCR with primers for the BRN2 promoter. The results (mean±SD of three determinations) are expressed as the relative-fold enrichment compared to that obtained with the IgG control (assigned a value of 1). **f** DU-145/tet-CshRNA (blue circles) and DU-145/tet-MUC1shRNA (red squares) cells seeded at $1 \times 10^4$ cells/ml were treated with vehicle (open symbols) or 500 ng/ml DOX (closed symbols) for the indicated times. Cell number (mean±SD of three replicates) was determined by trypan blue staining. **g** DU-145/tet-MUC1shRNA cells treated with vehicle or 500 ng/ml DOX for 7 days were assayed for invasive capacity in matrigel coated transwell chambers. Results (mean±SD of five determinations) are expressed as the relative invasive capacity compared to that obtained with the control cells (assigned a value of 1). **h** DU-145/tet-CshRNA and DU-145/tet-MUC1shRNA cells seeded at 100 cells/well in six-well plates were treated with vehicle or 500 ng/ml DOX. Colonies were stained with crystal violet on day 9. The results are expressed as the colony number (mean±SD of four determinations) per well. *$p < 0.05$ (Student's $t$-test). Source data are provided as a Source Data file.

and (ii) ASPC1, AURKA and SYP expression (Fig. 6b, left). Silencing MUC1-C was also associated with suppression of MYC, SOX2, KLF4 and OCT4 (Fig. 6b, right), supporting the association of MUC1-C signaling with induction of the OSKM pluripotency factors and LNCaP-AI tumorigenicity. The MUC1-C cytoplasmic domain (CD) includes a CQC motif that is an Achilles' heel for targeting MUC1-C function[14] (Fig. 6c). Cell-penetrating peptides, such as GO-201 and GO-203, that selectively target the MUC1-C CQC motif are effective in blocking MUC1-C homodimerization and nuclear localization[41–44] (Fig. 6c). MUC1-C peptide inhibitors were first evaluated in human PC xenograft models using GO-201 and CP-1, an identical control peptide with the exception that the critical CQC motif is mutated to AQA[41]. GO-201 was shown to be effective against human prostate, breast, pancreatic and esophageal squamous cell carcinoma tumor xenograft models at different dose-dependent schedules, whereas CP-1 had no apparent effect on tumor growth or histology[30,41,45,46]. GO-201 and GO-203 both contain the CQCRRKN sequence, block MUC1-C dimerization and have similar dose-dependent activity in vitro and in vivo[28,42,47,48]. In addition, treatment of tumors with GO-203 encapsulated in nanoparticles (GO-203/NPs) has demonstrated dose-dependent activity[49]. Here, in vitro treatment of LNCaP-AI

cells with GO-203 was associated with suppression of growth (Fig. 6d) and downregulation of (i) the MUC1→BRN2 pathway, (ii) MYCN, EZH2 and NE marker expression, and (iii) pluripotency factors (Fig. 6e, left and right). These findings were supported by GO-203/NP treatment of LNCaP-AI tumors growing in nude mice, which also resulted in inhibition of growth (Fig. 6f), decreases in expression of MUC1-C-driven NE markers and OSKM effectors of pluripotency (Fig. 6g, left and right).

In extending these studies, we found that treatment of DU-145 cells with GO-203 in vitro is similarly associated with inhibition of growth (Fig. 7a) and suppression of MUC1-C-induced NE markers and pluripotency factors (Fig. 7b, left and right). Moreover, GO-203/NP treatment of DU-145 xenografts resulted in suppression of tumorigenicity (Fig. 7c) and MUC1-C-induced signaling (Fig. 7d, left and right), confirming the findings in the LNCaP-AI model. These responses to targeting MUC1-C occurred in the absence of apparent changes in morphology, which may require longer periods of treatment for differentiation of that phenotypic characteristic. We also studied NCI-H660 NEPC cells, which constitutively express MUC1-C and BRN2 at higher levels than that found in DU-145 cells (Fig. 7e). Consistent with the LNCaP-AI and DU-145 models, targeting MUC1-C with GO-203 resulted in inhibition of NCI-H660 cell growth (Fig. 7f)

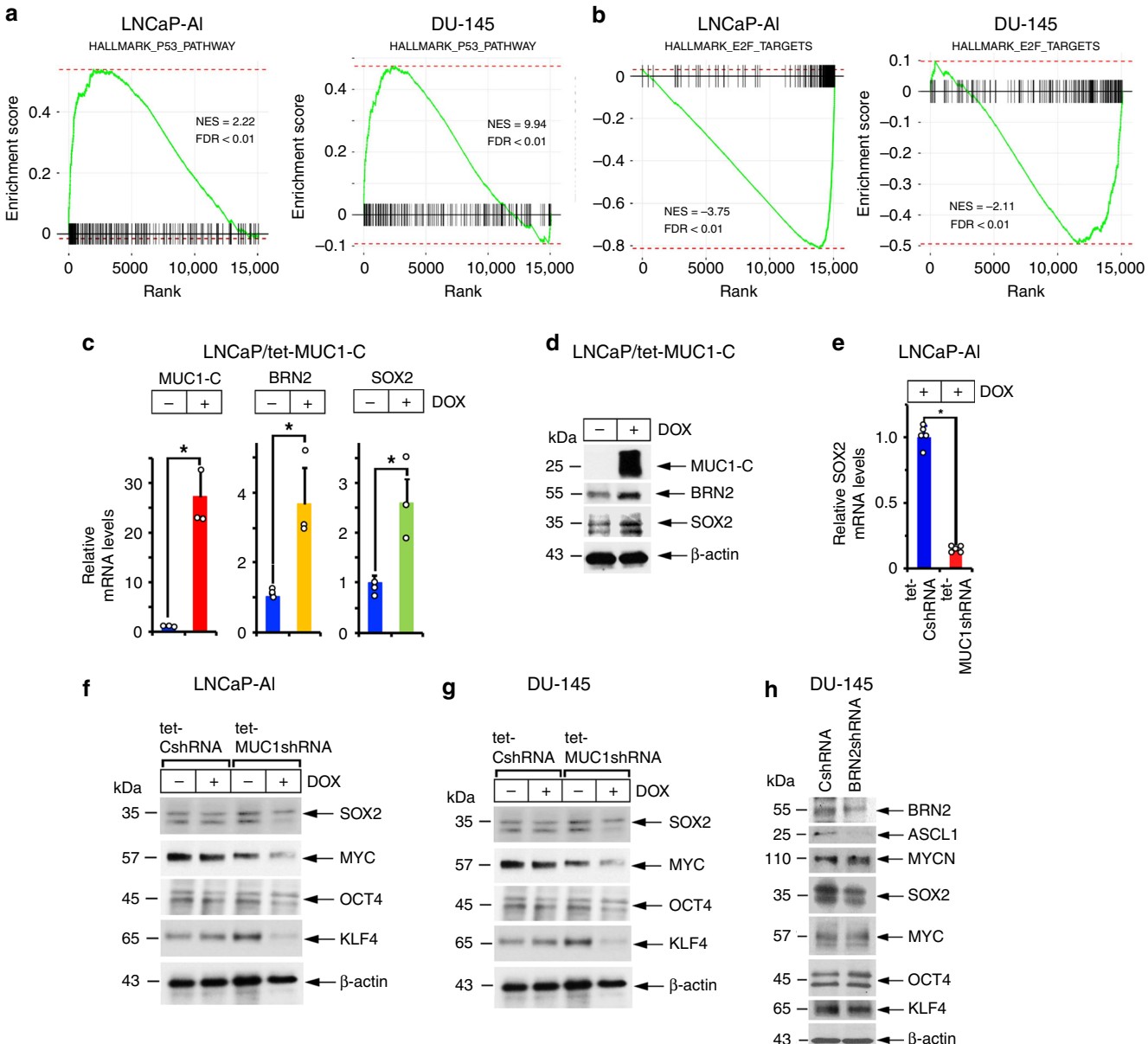

**Fig. 5 MUC1-C drives effectors of lineage plasticity. a** RNA-seq was performed in triplicate on LNCaP-AI/tet-MUC1shRNA (left) and DU-145/tet-MUC1shRNA (right) cells treated with vehicle or 500 ng/ml DOX for 7 days. The datasets were analyzed with GSEA, using the Hallmark gene signature collection for the p53 Pathway. **b** RNA-seq was performed in triplicate on LNCaP-AI/tet-MUC1shRNA (left) and DU-145/tet-MUC1shRNA (right) cells treated with vehicle or 500 ng/ml DOX for 7 days. The datasets were analyzed with GSEA, using the Hallmark gene signature collection for E2F Targets. **c** and **d**. LNCaP cells expressing tet-MUC1-C were treated with vehicle or 500 ng/ml DOX for 7 days. MUC1-C, BRN2 and SOX2 mRNA levels were analyzed by qRT-PCR (**c**). The results (mean±SD of three determinations) are expressed as relative mRNA levels compared to that obtained for vehicle-treated cells (assigned a value of 1). Lysates were immunoblotted with antibodies against the indicated proteins (**d**). **e** and **f** LNCaP-AI cells stably expressing a tet-CshRNA or tet-MUC1shRNA were treated with vehicle or 500 ng/ml DOX for 7 days. SOX2 mRNA levels were analyzed by qRT-PCR (**e**). The results (mean±SD of five determinations) are expressed as relative mRNA levels compared to that obtained for DOX-treated LNCaP-AI/tet-CshRNA cells (assigned a value of 1). Lysates were immunoblotted with the indicated antibodies (**f**). **g** Lysates from DU-145/tet-CshRNA and DU-145/tet-MUC1shRNA cells treated with vehicle or 500 ng/ml DOX for 7 days were immunoblotted with antibodies against the indicated proteins. **h** Lysates from DU-145/CshRNA and DU-145/BRN2shRNA cells were immunoblotted with antibodies against the indicated proteins. *$p < 0.05$ (Student's $t$-test). Source data are provided as a Source Data file.

and suppression of MUC1-C signaling pathways linked to NE differentiation (Fig. 7g, left) and pluripotency (Fig. 7g, right).

**MUC1 expression correlates with BRN2 and the NEPC score.** In extending these findings to PC tissues, we found that *MUC1* is amplified in 29.9% (32/107) of a NEPC enriched CRPC cohort[9],

compared to 6.0% (9/150) in the SU2C CRPC cohort with minimal NEPC[50] and 1.8% (6/333) in the TCGA primary prostate adenocarcinoma cohort[51] (Fig. 8a). MUC1 expression was also significantly increased in CRPCs compared to localized, hormone-naïve PCs (Fig. 8b). Further analysis showed that MUC1 high CRPC tumors associate with decreased AR, KLK3 TMPRSS2, HERC3, and NKX3-1 expression levels (Fig. 8c).

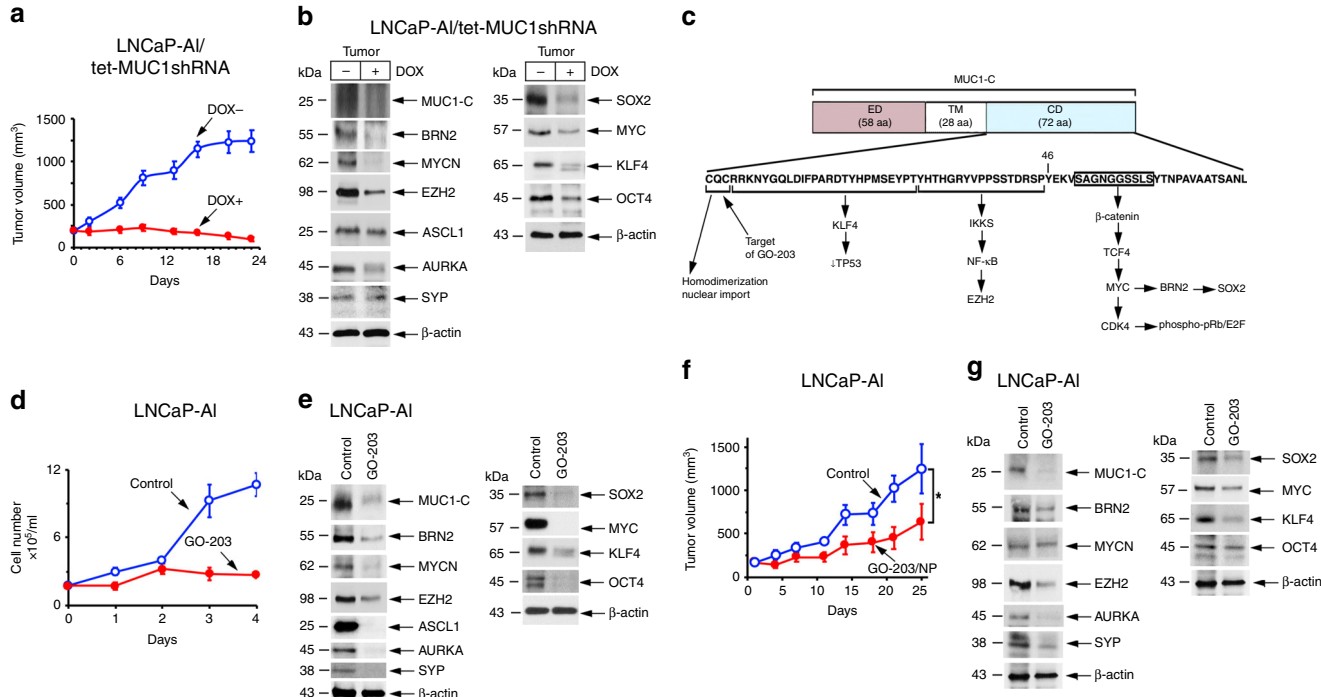

**Fig. 6 Targeting MUC1-C suppresses LNCaP-AI tumorigenicity. a** Six-week old nude male mice were injected subcutaneously in the flank with $3 \times 10^6$ LNCaP-AI/tet-MUC1shRNA cells. Mice were pair-matched into two groups when tumors reached 100–150 mm³ and were fed without and with DOX. Tumor volumes are expressed as the mean±SD for six mice. **b** Lysates from tumors obtained on day 24 were immunoblotted with antibodies against the indicated proteins. **c** Schema of the MUC1-C subunit with the amino acid sequence of the 72 aa cytoplasmic domain. Highlighted is the CQC motif, which is necessary for MUC1-C homodimerization, nuclear localization and function. The CQC motif is targeted by the cell-penetrating GO-203 peptide ((R₉)-CQCRRKN). Highlighted is the MUC1-C→IKK→NF-κB p65 pathway that activates *EZH2* intron 1. Also highlighted is the region of the MUC1-C cytoplasmic domain that binds directly to β-catenin and TCF4 and induces *MYC* transcription[28,29]. In turn, MYC activates CDK4 with phosphorylation and inhibition of RB[28,35]. The MUC1-C cytoplasmic domain also functions in regulating p53 expression and function[22]. **d** LNCaP-AI cells seeded at $2 \times 10^4$ cells/ml in androgen-depleted medium were left untreated or treated with 5 μM GO-203 for the indicated days. Cell number (mean±SD of three determinations) was determined by trypan blue staining. **e** Lysates obtained on day 3 were immunoblotted with antibodies against the indicated proteins. **f** Castrated 6-week old nude male mice were injected subcutaneously in the flank with $3 \times 10^6$ LNCaP-AI cells. Mice were pair-matched into two groups when tumors reached 100–150 mm³ and were treated IV with vehicle or GO-203/NPs weekly for 3 weeks. Tumor volumes are expressed as the mean±SD for six mice. **g** Lysates from tumors obtained on day 25 were immunoblotted with antibodies against the indicated proteins. Source data are provided as a Source Data file.

These studies were extended by analysis of a comprehensive genomic and transcriptomic dataset generated from 429 patients with advanced prostate cancer[4]. The results showed that MUC1 expression significantly associates with decreases in PSA/KLK3 (Fig. 8d) and increases in BRN2 (Fig. 8e), providing support for the central premise that MUC1-C is upregulated in PC progression in association with downregulation of AR signaling and induction of the BRN2 pathway. BRN2 overexpression in CRPC cells drives SOX2, induces NE markers and enriches for an NEPC gene signature[7]. Here, we found that MUC1 is also significantly associated with SOX2 expression (Fig. 8f) and the NEPC score (Fig. 8g), an RNA-based NE expression signature[4], supporting a MUC1-C→BRN2→SOX2 pathway in driving NE differentiation in advanced prostate cancers.

## Discussion

NEPC is a lethal form of PC that is increasing in incidence in association with the development of resistance to AR pathway inhibitors[1,2,4,52,53]. The limited options for treating patients with de novo or treatment-related NEPC have emphasized the need for identifying druggable targets. The present studies uncover a previously unrecognized role for the MUC1-C oncoprotein in driving the lineage plasticity of PC to CRPC and NEPC. Evidence in support of MUC1-C functioning in lineage switching emerged in part with the generation of a model in which AR-dependent

LNCaP cells were selected for growth under androgen-depleted conditions. The resulting LNCaP-AI cells were found to have upregulation of MUC1-C expression in association with suppression of AR axis signaling. In addition, findings that MUC1-C contributes to induction of (i) the BRN2 neural TF[39], and (ii) MYCN, EZH2 and selected NE markers (ASCL1, AUROKA and SYP), which have been associated with progression to NEPC[8,12], provided further support for the notion that MUC1-C drives NE differentiation in PC. To address a potential concern that the effects of MUC1-C on lineage switching are limited to the LNCaP cell model, we studied DU-145 cells isolated from a patient with metastatic CRPC[31]. Unlike LNCaP cells, DU-145 cells constitutively express MUC1-C in the presence of low to undetectable levels of AR, consistent with an inverse relationship between MUC1 and AR in PC cell lines. Silencing of MUC1-C in this model also resulted in suppression of BRN2, MYCN, EZH2, and NE markers, suggesting that targeting MUC1-C represents an approach for attenuating progression to the NE lineage. To further address that contention, experiments were performed with NCI-H660 cells, which were derived from a patient with NEPC and have high levels of MUC1-C expression. Targeting MUC1-C in this model also suppressed BRN2, MYCN, EZH2, and NE markers, consistent with the premise that MUC1-C is sufficient to drive lineage plasticity of PC to CRPC-NE and NEPC.

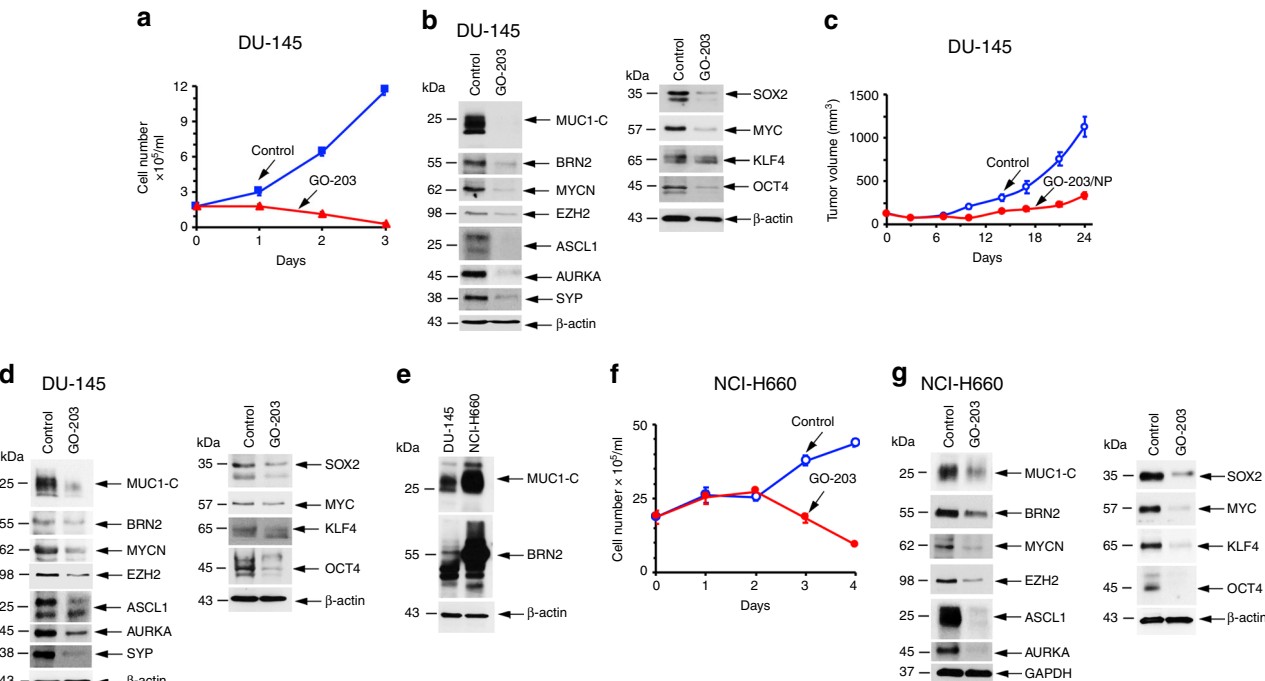

**Fig. 7 Targeting MUC1-C suppresses BRN2 and self-renewal. a** DU-145 cells seeded at $5 \times 10^4$ cells/ml in androgen-depleted medium were left untreated or treated with 5 μM GO-203 for the indicated times. Cell number (mean±SD of three determinations) was determined by trypan blue staining. **b** Lysates obtained on day 3 were immunoblotted with antibodies against the indicated proteins. **c** Castrated 6-week old nude male mice were injected subcutaneously in the flank with $3 \times 10^6$ DU-145 cells. Mice were pair-matched into two groups when tumors reached 100–150 mm³ and were treated IV with vehicle or GO-203/NPs weekly for 3 weeks. Tumor volumes are expressed as the mean±SD for six mice. **d** Lysates from tumors obtained on day 25 were immunoblotted with antibodies against the indicated proteins. **e** Lysates from DU-145 and NCI-H660 cells were immunoblotted with antibodies against the indicated proteins. **f** NCI-H660 cells seeded at $1 \times 10^6$ cells/ml in androgen-depleted medium were left untreated or treated with 5 μM GO-203 for the indicated times. Cell number (mean±SD of three determinations) was determined by trypan blue staining. **g** Lysates obtained on day 2 were immunoblotted with antibodies against the indicated proteins. Source data are provided as a Source Data file.

A reciprocal interaction between AR and MUC1-C was first identified with the demonstration that AR occupies the *MUC1* promoter and represses *MUC1* transcription in LNCaP cells[54]. A negative interplay between AR and MUC1-C in PC cells was further supported by the finding that enforced upregulation of MUC1-C is associated with suppression of AR axis signaling[55]. The present work provides insights into the role of MUC1-C in PC progression by showing that MUC1-C drives the *BRN2* gene. AR represses *BRN2* activation[7]. Therefore, MUC1-C-mediated downregulation of AR expression and/or transactivation function provided a mechanistic explanation for MUC1-C indirectly contributing to induction of BRN2 mRNA and protein. However, the observation that MUC1-C occupies the *BRN2* promoter invoked the possibility for a direct effect. MUC1-C activates the inflammatory TAK1→IKK→NF-κB p65 pathway and, by binding directly to NF-κB p65, promotes activation of NF-κB p65 target genes, including (i) ZEB1 and thereby EMT, (ii) EZH2 with increases in H3K27me3, and (iii) DNMT1/3b with alterations in DNA methylation patterns[16] (Fig. 9). By extension, the induction of EMT and upregulation of EZH2 and DNMTs have been associated with progression to NEPC[9,56]. In the present studies, targeting NF-κB p65 genetically or with the BAY-11-7085 inhibitor had no effect on BRN2 expression. In addition to NF-κB, MUC1-C activates the MYC pathway, binds directly to the MYC HLH/LZ domain and promotes occupancy of MYC on its target genes[23,28,29]. In this respect, we identified an E-box as a MYC binding site that functions in activating the *BRN2* promoter. Moreover, we found that (i) MUC1-C and MYC are detectable on the *BRN2* promoter, and (ii) targeting MUC1-C→MYC signaling results in suppression of *BRN2* promoter activation and BRN2

expression. These findings collectively supported at least two mechanisms for MUC1-C induced BRN2 expression; that is indirectly by repression of AR signaling and directly by MYC-mediated *BRN2* activation (Fig. 9).

NEPC emerges with resistance to AR pathway inhibitors and is associated with activation of gene programs that confer EMT, the CSC state and NE differentiation[1,2,52,53]. In this regard, MUC1-C drives EMT, epigenetic reprogramming and the capacity for self-renewal of human breast and lung cancer cells[15,16]. MUC1-C also suppresses the p53 and RB pathways[22,28,35,57], which cooperate to suppress PC lineage plasticity and anti-androgen resistance[34]. Here, in the LNCaP-AI and DU-145 cell models, MUC1-C induced gene expression patterns were significantly associated with (i) downregulation of the AR response, (ii) suppression p53 signaling, and (iii) activation of the E2F Targets gene set in support of regulating the RB-E2F axis[37]. In addition, we found that MUC1-C induces the MYC→BRN2→SOX2 pathway, invasion, self-renewal and tumorigenicity, supporting a role for MUC1-C in conferring PC stemness. Having shown that MUC1-C drives MYC and SOX2, we also found MUC1-C induction of KLF4 and OCT4, which collectively as the Yamanaka OSKM pluripotency reprogramming factors confer lineage plasticity and dedifferentiation of fibroblasts[40]. How MUC1-C induces KLF4 and OCT4 will require subsequent investigation. Nonetheless, these findings are of potential relevance for MUC1-C-induced PC progression in that suppression of the p53 and RB pathways, which can function as repressors of pluripotency, enhances efficiency of the OSKM pluripotency factors in inducing lineage plasticity of somatic cells[40,58]. Lineage plasticity in cancer includes in part dedifferentiation with the reacquisition of stem

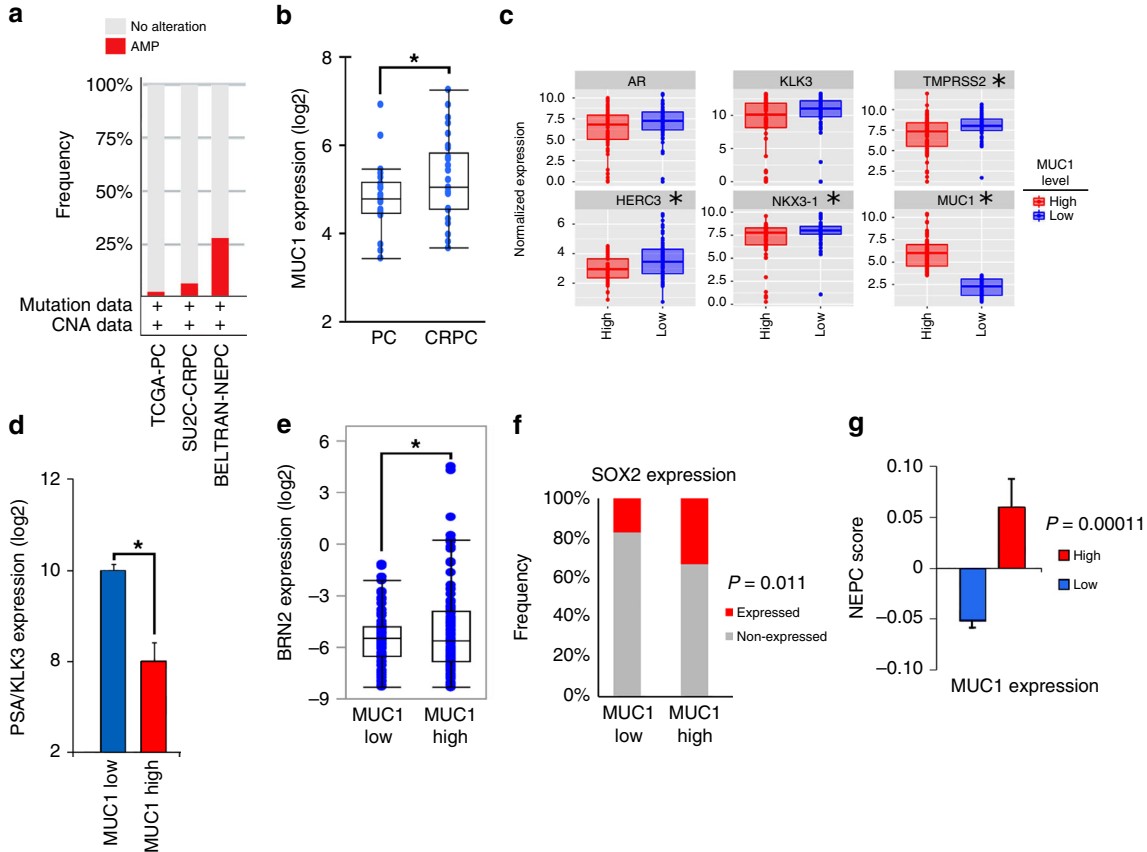

**Fig. 8 MUC1 overexpression associates with NE differentiation. a** *MUC1* copy-number alteration (CNA) data for the TCGA-PRAD[51], SU2C-CRPC[50], and NEPC[9] cohorts. **b** Localized prostate cancer, hormone-naïve samples (n = 22) were compared to metastatic CRPC samples (n = 29)[9,50]. Multiple probe set IDs for *MUC1* were averaged for each patient sample after normalization to obtain a representative expression value for the gene. The center line indicates the median value, bounds of the box denote 25th (lower) and 75th (upper) percentiles, and whiskers indicate minimum (lower) and maximum (upper) values excluding outliers. Student's *t*-test was used to compare groups (*p*-value = 0.038). **c** Normalized expression data for the SU2C-CRPC cohort were downloaded from cBioPortal, and median expression used to group samples into MUC1 high and MUC1 low groups. Expression of AR and AR target genes was assessed in MUC1 high and MUC1 low groups using a Wilcoxon rank-sum test. Boxplots represent the 1st quartile, median and 3rd quartile of each distribution. Whiskers extend to the maximum and minimum values up to 1.5*interquartile range (IQR). **d**–**g** Data were downloaded from cbioportal[4]. NEPCs and CRPCs were analyzed together. **d** Samples were dichotomized by *MUC1* high (n = 105) and *MUC1* low (n = 106) expression defined by the normalized median expression value. Samples were analyzed for *KLK3* expression. Student's *t*-test was used to compare groups (*p* < 0.0001). **e** Samples were dichotomized by *MUC1* high (n = 116) and *MUC1* low (n = 96) expression defined as normalized expression value ≥ 1.4 or < 1.4. Samples with undetectable *BRN2* values were excluded. Student's *t*-test was used to compare groups (*p* = 0.038). **f** Samples were dichotomized by *MUC1* high (n = 106) and *MUC1* low (n = 106) expression defined as normalized median expression value. Presence of SOX2 expression was defined as an FKPM value > 0. Fisher's exact test was used to compare groups. **g** Samples were dichotomized by *MUC1* high (n = 80) and *MUC1* low (n = 66) expression defined as normalized expression value. Samples with NEPC values were retained for analysis. NEPC score was calculated using polyA RNA-seq data as described[4]. Student's *t*-test was used to compare groups (*p* = 0.0001).

cell features[2,59,60]. Stemness contributes to cancer progression and treatment resistance, and thus understanding how cancer cells acquire plasticity is of critical importance. To our knowledge, the present work supports a previously unrecognized role for MUC1-C in driving dedifferentiation of PC cells (Fig. 9). Lineage plasticity has also been used to describe transdifferentiation, which involves fate switching to another differentiated cell type[2,59,60]. Using this definition, the present results lend further support to a role for MUC1-C in promoting transdifferentiation of androgen-dependent PC cells to androgen-independent PC cells with NE features (Fig. 9). Along these lines, additional investigation will be needed to more precisely define whether MUC1-C contributes to phenotypic plasticity of PC cells by driving pluripotency, dedifferentiation, transdifferentiation or the interconnectivity between these states[61].

Ectopic expression of MUC1 in 3Y1 fibroblasts was found to be sufficient to induce anchorage-independent growth and tumorigenicity, consistent with an oncogenic function[62]. Subsequent work demonstrated that MUC1-C is an oncoprotein, which confers multiple hallmarks of the cancer cell, including EMT, the CSC state, epigenetic reprogramming, drug resistance and immune evasion[15,16]. Clearly, the upregulation of MUC1 expression per se is not a transforming event. Along these lines, MUC1 is highly expressed in the lactating mammary gland, which (i) occurs in association with suppression of the EMT program to preserve epithelial integrity and differentiation[63,64] and (ii) upon remodeling during involution rarely progresses to breast cancer. In cancer cells, MUC1-C is upregulated by auto-inductive circuits resulting from interactions with proinflammatory TFs, such as NF-κB p65 and STAT3, which are activated by stress and drive the EMT program[65,66]. These findings provide support for the notion that MUC1-C contributes to cancer progression, at least in part, in association with the response to stress and inflammation, activation of an EMT

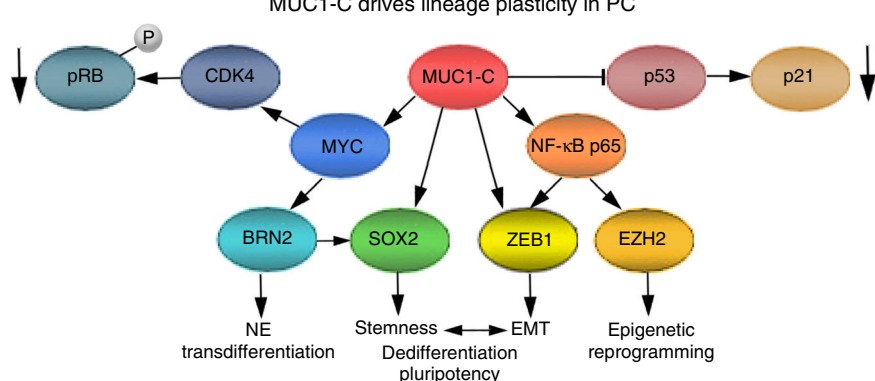

**Fig. 9 Proposed model for MUC1-C in driving PC lineage plasticity.** MUC1-C activates the *MYC* gene, binds directly to the MYC HLH/LZ domain and contributes to induction of MYC target genes, including *CDK4* with phosphorylation of RB[28,29,35]. MUC1-C also promotes inactivation of RB by MYC/BMI1-mediated suppression of CDKN2A/p16 (ref. [36]). In the present studies, we found that MUC1-C induces MYC occupancy on the *BRN2* promoter with induction of BRN2 expression. In turn, BRN2 induces SOX2 expression[7]. In addition to MYC and SOX2, we show that MUC1-C drives KLF4 and OCT4 (OSKM), which are collectively sufficient for inducing pluripotency and dedifferentiation of somatic cells[40]. MUC1-C also suppresses the p53 signaling pathway and CDKN1A/p21. In addition to MYC, MUC1-C activates the inflammatory TAK1→IKK→NF-κB p65 pathway and, by binding directly to NF-κB p65, promotes activation of NF-κB p65 target genes[65], including (i) ZEB1 and thereby EMT and stemness, and (ii) EZH2 with increases in H3K27me3 (refs. [16,35]). In this way, MUC1-C integrates activation of the MYC and NF-κB p65 pathways with suppression of p53 and regulation of the RB-E2F axis to drive PC lineage plasticity with dedifferentiation and pluripotency.

program and induction of pluripotency, as would occur in non-cancerous cells to promote wound healing and maintain tissue homeostasis[15,16,66]. Viewed in this way and given the association of prostatitis with EMT[67,68] and prostate cancer[69], prolonged activation of MUC1-C in settings of chronic inflammation and, in turn, the induction of stemness and reprogramming could hold important implications for PC progression and treatment. In this respect and of potential clinical relevance, targeting MUC1-C in PC cells with the GO-203 inhibitor, which blocks MUC1-C homodimerization and nuclear localization[42–44], phenocopied the effects of MUC1-C silencing, including downregulation of (i) MYC→BRN2 signaling, (ii) MYCN, EZH2 and NE markers, and (iii) the OSKM pluripotency factors. Treatment with GO-203 was also associated with inhibition of self-renewal and tumorigenicity, indicating that this agent is active in targeting MUC1-C-induced stemness. GO-203 has been evaluated in early phase clinical trials and, based on an acceptable safety profile and evidence of anti-tumor activity, is being further developed to target MUC1-C expressing cancers, such as CRPC and NEPC. The short half-life of GO-203 necessitated daily intravenous delivery, which is a challenging schedule in the clinic. Accordingly, GO-203 has been encapsulated in polymeric NPs (GO-203/NPs) and, based on the dose-dependent anti-tumor activity of this formulation, is under development for more convenient weekly administration[49]. Other work has demonstrated that the MUC1-C extracellular domain is druggable with antibody-based approaches, including antibody-drug conjugates (ADCs) and potentially chimeric antigen receptors[70]. The present findings lend support for the contention that these MUC1-C-targeted agents could enable therapeutic strategies for attenuating PC progression to NEPC and for treating this aggressive disease.

## Methods

**Cell culture.** Human LNCaP (ATCC), C4-2B[25] and DU-145 (ATCC) cells were cultured in RPMI1640 medium (Corning Life Sciences, Corning, NY, USA) containing 10% heat-inactivated fetal bovine serum (FBS; GEMINI Bio-Products, West Sacramento, CA, USA). LNCaP-AI cells were established by culturing C4-2B cells in phenol red-free RPMI1640 medium (Thermo Fisher Scientific, Waltham, MA, USA) containing 10% charcoal-stripped FBS (Millipore Sigma, Burlington, MA, USA) for over 6 months. Human NCI-H660 NEPC cells (ATCC) were cultured in RPMI1640 medium with 5% FBS, 10 nM β-estradiol (Millipore Sigma), 10 nM hydrocortisone, 1% insulin-transferrin-selenium (Thermo Fisher Scientific) and

2 mM L-glutamine (Thermo Fisher Scientific). Cells were treated with the MUC1-C inhibitor GO-203 (refs [42–44]), the AR pathway inhibitor enzalutamide (ENZ; Santa Cruz Biotechnology, Dallas, TX, USA) and the BET bromodomain inhibitor JQ1. Cell growth and viability were assessed by 0.4% trypan blue (Thermo Fisher Scientific) exclusion. Authentication of the cells was performed by short tandem repeat (STR) analysis. Cells were monitored for mycoplasma contamination using the MycoAlert Mycoplasma Detection Kit (Lonza, Rockland, ME, USA).

**Tetracycline-inducible gene silencing.** MUC1shRNA (MISSION shRNA TRCN0000122938; Sigma), MYCshRNA (MISSION shRNA TRCN0000039642; Sigma) or a control scrambled shRNA (CshRNA; Sigma) was inserted into the pLKO-tet-puro vector (Plasmid #21915; Addgene, Cambridge, MA, USA). BRN2shRNA (MISSION shRNA TRCN0000019330; Sigma) was inserted into the pLKO-puro vector. The viral vectors were produced in 293T cells[35]. Cells transduced with the vectors were selected for growth in 1–3 μg/ml puromycin. For tet-inducible vectors, cells were treated with 0.1% DMSO as the vehicle control or doxycycline (DOX; Millipore Sigma).

**Quantitative reverse-transcription PCR (qRT-PCR).** Total cellular RNA was isolated using Trizol reagent (Thermo Fisher Scientific). cDNAs were synthesized using the High Capacity cDNA Reverse Transcription Kit (Applied Biosystems, Grand Island, NY, USA). The cDNA samples were amplified using the Power SYBR Green PCR Master Mix (Applied Biosystems) and the CFX96 Real-Time PCR System (BIO-RAD, Hercules, CA, USA)[24]. Primers used for qRT-PCR are listed in Supplementary Table 1.

**Immunoblotting.** Total lysates prepared from subconfluent cells as described[24] were subjected to immunoblot analysis using anti-AR (H-280, 1:100 dilution; Santa Cruz Biotechnology), anti-PSA (5365, 1:1000 dilution; Cell Signaling Technology, Danvers, MA, USA), anti-NKX3.1 (83700, 1:1000 dilution; Cell Signaling Technology), anti-β-actin (1:100,000 dilution; Sigma), anti-MUC1-C (HM-1630-P1ABX, 1:400 dilution; Thermo Fisher Scientific, Waltham, MA, USA), anti-EZH2 (5246, 1:1000 dilution; Cell Signaling Technology), anti-MYCN (9405, 1:1000 dilution; Cell Signaling Technology), anti-BRN2 (12137, 1:1000 dilution; Cell Signaling Technology), anti-MYC (ab32072, 1:1000 dilution; Abcam, Cambridge, MA), anti-SOX2 (3579, 1:1000 dilution; Cell Signaling Technology), anti-ASCL1 (GTX129189, 1:1000 dilution; GeneTex, Irvine, CA, USA), anti-AUROKA (ab1287, 1:4000 dilution; Abcam), anti-SYP (MA5-16402, 1:200 dilution; Thermo Fisher Scientific), anti-KLF4 (12173, 1:1000 dilution; Cell Signaling Technology) and anti-OCT4 (2750, 1:1000 dilution; Cell Signaling Technology).

**Chromatin immunoprecipitation (ChIP) assays.** Soluble chromatin was precipitated with anti-MUC1-C (HM-1630-P1ABX, 1:50 dilution), anti-AR (H-280, 1:50 dilution), anti-MYC (ab56, 1:50 dilution; Abcam) or a control non-immune IgG (Santa Cruz Biotechnology). The precipitates were analyzed by qPCR using the Power SYBR Green PCR Master Mix and the ABI Prism 7300 sequence detector (Applied Biosystems). Data are reported as relative-fold enrichment[24]. Primers used for ChIP qPCR are listed in Supplementary Table 2.

**Invasion assays**. Cell invasion assays were performed in transwell chambers (3406; Sigma) precoated with matrigel[66].

**Colony formation assays**. Cells were seeded in 6-well plates for 24 h and then treated with DOX every 4 days. After 9–14 days, the cells were stained with 0.5% crystal violet (LabChem, Zelienople, PA, USA) in 25% methanol. Colonies >25 cells were counted in triplicate wells.

**Tumorsphere formation assays**. Cells ($5 \times 10^3$) were seeded per well in 6-well ultra-low attachment culture plates (Corning Life Sciences) in DMEM/F12 50/50 medium (Corning Life Sciences) with 20 ng/ml EGF (Millipore Sigma), 20 ng/ml bFGF (Millipore Sigma) and 1% B27 supplement (Gibco). Cells were treated with vehicle or 500 ng/ml DOX for 10–14 days. Tumorspheres were counted under an inverted microscope in triplicate wells.

**Promoter-reporter assays**. Cells were transfected with (i) an empty pGL3-basic vector, (ii) pBRN2-Luc (pGL410-BRN2p, Plasmid #110733; Addgene), (iii) pBRN2-Luc in which the E-box element CAAGTG at position −575 to −570 was mutated to AAAGCC (MUT1), (iv) pBRN2-Luc in which the E-box element CAGATG at position −511 to −506 was mutated to AAGACC (MUT2), and (v) SV-40-Renilla-Luc in the presence of Lipofectamine 3000 Reagent (Invitrogen). At 48 h after transfection, cells were lysed using passive lysis buffer (Promega, Madison, WI, USA). Luminescence was detected with the Dual-Luciferase Reporter Assay System (Promega).

**Mouse tumor model studies**. Six- to 8-week old male nude mice (Taconic Farms, Germantown, NY, USA) were injected subcutaneously in the flank with $3 \times 10^6$ tumor cells in 100 µl of a 1:1 solution of medium and Matrigel (BD Biosciences). In certain studies, the mice were castrated at least 3 days before cell inoculation. When the mean tumor volume reached 100–150 mm³, mice were pair-matched into groups and (i) treated with vehicle or GO-203/NPs (15 mg/kg IV weekly)[49] or (ii) fed without or with DOX (625 ppm, daily). Tumor measurements and body weights were recorded twice each week. Mice were sacrificed when tumors reached >1000 mm³ as calculated by the formula: (width)² × length/2. These studies were conducted in accordance with ethical regulations required for approval by the Dana-Farber Cancer Institute Animal Care and Use Committee (IACUC) under protocol 03-029.

**RNA-seq analysis**. Total RNA from cells cultured in triplicates was isolated using Trizol reagent (Invitrogen). TruSeq Stranded mRNA (Illumina, San Diego, CA, USA) was used for library preparation. Raw sequencing reads were aligned to the human genome (GRCh38.74) using STAR. Raw feature counts were normalized and differential expression analysis using DESeq2. Differential expression rank order was utilized for subsequent Gene Set Enrichment Analysis (GSEA), performed using the fgsea (v1.8.0) package in R. Gene sets queried included those from the Hallmark Gene Sets available through the Molecular Signatures Database (MSigDB).

**Statistical analysis**. Each experiment was performed at least three times from distinct samples. Data are expressed as the mean ± SD. The unpaired Mann–Whitney $U$ test and Student's $t$-test were used to determine differences between means of groups. A $p$-value of <0.05 denoted by an asterisk (*) was considered statistically significant.

**Analysis of human PC datasets**. Data were pre-processed, RMA-normalized, and log2-transformed[4,65]. A quantile-quantile plot was used to assess for data normality. Data analysis was performed using the cBioPortal Cancer Genomic and Oncomine websites[9,50]. GSE32269 was downloaded from Gene Expression Omnibus (GEO).

**Reporting summary**. Further information on research design is available in the Nature Research Reporting Summary linked to this article.

## Data availability

The RNA-seq data have been deposited in the GEO database under accession code GSE139335. The TCGA-PRAD, SU2C-CRPC and NEPC cohorts referenced during the study are available from the cBioPortal (http://www.cbioportal.org/ and https://www.cbioportal.org/study/summary?id=prad_su2c_2019) and from the GEO database under accession code GSE32269. The source data underlying Figs. 1–6 and Supplementary Figs. 1–4 are provided as a Source Data file. All other data supporting the findings of this study are available within the article and its supplementary information files and from the corresponding author upon reasonable request. A reporting summary for this article is available as a Supplementary Information file.

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

## Acknowledgements

The C4-2B cell line was kindly provided by Dr. Li Jia, Department of Surgery, Brigham and Women's Hospital, Harvard Medical School, Boston, MA, USA. Research reported in this publication was supported by the National Cancer Institute of the National Institutes of Health under grant numbers CA97098, CA166480, CA229716 and CA233084 awarded to D.K. and CA232979 awarded to S.L.

## Author contributions

Conceptualization, Y.Y., K.K.W., M.O. and D.K.; Methodology, Y.Y., H.R., C.J., T.H., S.P., M.L., Q.H., S.L., L.K. and M.S.; Investigation, Y.Y., H.R., C.J., T.H., M.H., W.L., N.Y., A.F., M.Y., Y.Z., N.Z., D.H., T.M. and T.K.; Bioinformatics Analysis, S.P., M.L., Q.H., S.L., L.K., M.S.; Writing-Original Draft, D.K.; Writing-Review and Editing, Y.Y., H.R., S.L., S.P., K.K.W. and D.K.; Funding Acquisition, T.K., K.K.W., M.O., L.S. and D.K.; Resources, M.L., Q.H., S.L., S.P. and M.S.; Supervision, T.K., K.K.W., M.O., L.S. and D.K.

## Competing interests

D.K. has equity interests in Genus Oncology, Reata Pharmaceuticals, Hillstream Bio-Pharma, Nanogen Therapeutics and Victa BioTherapeutics, serves as a member of the board of directors of Nanogen and Victa, and is a paid consultant to Reata, CanBas and Victa. The other authors declare no competing interests.
