## [Peer Review File · Nature Communications]

Reviewers' Comments:

Reviewer #1:

Remarks to the Author:

This manuscript investigates the oncogenic role of MUC1-C in neuroendocrine prostate cancer (NEPC). The study shows that MUC1-C is highly expressed in androgen-independent (AI) PC cells. Furthermore, the authors claim that MUC1-C upregulates BRN2 expression through a MYC-mediated pathway; it also regulates p53 signaling and MYCN and EZH2 expression. These pathways and factors induce prostate cancer cells into NEPC cells. The authors conclude that MUC1-C serves as a regulator of lineage plasticity in advanced PC. However, MUC1-C regulation of AR signaling, AR suppression of NE phenotype, MUC1-C activation of MYC, and the connection of BRN2 with NEPC are all known. Mechanistically, it is also unclear how MUC1-C facilitates MYC occupancy at the BRN2 promoter.

Other points:

1. C4-2B cells are CRPC cells (Thalmann GN, et al, Cancer Res, 1994). Why is MUC1-C not highly expressed in C4-2B cells, compared to LNCaP cells? Was the use of androgen-independent (AI) medium the only way to establish MUC1-C high expression cells? The authors should do a cell line-based survey to determine the relationship between MUC1-C and AR.
2. Morphologically, NEPC shares features with other high grade neuroendocrine cancers, including presence of small cells with 'salt and pepper' chromatin, high mitotic count and nuclear molding (Wang W et al, Am J Surg Pathol, 2008). The authors should perform cell morphology analysis and H&E staining in xenografts to demonstrate the progression of CRPC to NEPC.
3. In Figure 3, the authors demonstrated that MUC1-C and MYC occupy the same region in the BRN2's promoter. The evidence is too primary. The authors should mutate the putative MYC binding sites to examine MUC1-C and MYC binding by EMSA. They also should determine their occupancy on the same region using their antibodies in EMSA assay.
4. Importantly, whether or not MUC1-C induces NEPC through BRN2 is unclear. Both loss of function and gain of function as well as rescue experiments should be performed.
5. NEPC often links to tumor aggressiveness such as cell metastasis in patients. If MUC1-C affects the transition from CRPC to NEPC, the authors should focus on invasive and metastasis assay in cells and mice, but not only cell growth assay. Since the authors already performed the xenograft assay, IHC of NE markers should be examined in xenografts. How about cell morphology changes?
6. In Figure 7, the relationship between MUC1-C and BRN2 in hormone naïve PCa, CRPC and NEPC tissues should be examined systematically.

Reviewer #2:

None

The comments of the reviewers have been addressed as follows:

Reviewer #1

1. However, MUC1-C regulation of AR signaling, AR suppression of NE phenotype, MUC1-C activation of MYC, and the connection of BRN2 with NEPC are all known. Mechanistically, it is also unclear how MUC1-C facilitates MYC occupancy at the BRN2 promoter.

In response, MUC1-C had been linked to the suppression of AR signaling by unclear mechanisms, which have now been uncovered by the present work. Of further importance, MUC1-C has not been previously associated with driving BRN2 expression and neuroendocrine dedifferentiation in prostate or other cancers. These points have been emphasized in the Introduction (p. 5).

Regarding MYC, the MUC1-C cytoplasmic domain binds directly to the MYC HLH-LZ region, which plays an essential role in the MYC transactivation function. As a result, MUC1-C forms a complex with MYC on the promoters of MYC target genes and contributes to their activation. This information has been included in the Results (pp. 7-8) and Discussion (p. 13) sections.

2. C4-2B cells are CRPC cells (Thalman GN, et al, Cancer Res, 1994). Why is MUC1-C not highly expressed in C4-2B cells, compared to LNCaP cells? Was the use of androgen-independent (AI) medium the only way to establish MUC1-C high expression cells? The authors should do a cell line-based survey to determine the relationship between MUC1-C and AR.

As previously described and as shown in the present work (Figs. 1a and 1b), LNCaP and C4-2B cells (i) express AR and are active in AR axis signaling, and (ii) are dependent on androgen for growth *in vitro*. Selection of C4-2B cells for growth under androgen-depleted conditions was performed to determine whether MUC1-C plays a role in the androgen independent (AI) phenotype (Figs. 1c-1f). A statement to this effect has been included in the Results section (p. 6).

As requested, we analyzed the "Cancer Cell Line Encyclopedia" database and found an inverse relationship between MUC1 and AR in PC cell lines, consistent with a role for MUC1-C in driving androgen-independent progression (Results, p. 8; new Supplementary Fig. 5).

3. Morphologically, NEPC shares features with other high grade neuroendocrine cancers, including presence of small cells with 'salt and pepper' chromatin, high mitotic count and nuclear molding (Wang W et al, Am J Surg Pathol, 2008). The authors should perform cell morphology analysis and H&E staining in xenografts to demonstrate the progression of CRPC to NEPC.

In response and as examined by phase contrast microscopy, the LNCaP-AI cells exhibit distinct patterns of growth with the formation of clusters compared to that found for C4-2B cells (Supplementary Fig. 3a). Staining with H&E further demonstrated that C4-2B cells have dense round or oval nuclei with diffuse chromatin and the absence of distinct nucleoli (Supplementary Fig. 3b, left panels). In contrast, LNCaP-AI cells were found to have larger irregular nuclei, visible nucleoli and occasional giant cells with smudgy chromatin, consistent in part with morphologic features identified in certain small cell carcinomas of the prostate (Supplementary Fig. 3b, right panels). These findings have been described in the Results section (p. 6).

Regarding studies in xenografts, we show that targeting MUC1-C genetically and pharmacologically suppresses expression of BRN2 and NE markers in association with decreases in tumorigenicity. These effects occur in the absence of apparent morphologic changes, which may require longer periods of treatment for differentiation of that phenotypic characteristic. A statement to this effect has been included in the Results section (p. 10).

4. In Figure 3, the authors demonstrated that MUC1-C and MYC occupy the same region in the BRN2's promoter. The evidence is too primary. The authors should mutate the putative MYC binding sites to examine MUC1-C and MYC binding by EMSA. They also should determine their occupancy on the same region using their antibodies in EMSA assay.

As requested, we have confirmed by re-ChIP analysis that MUC1-C and MYC form a complex on the BRN2 promoter (new Figs. 3c, right and 4c, right). Functional studies, in lieu of EMSAs, performed with a BRN2 promoter-luciferase reporter (pBRN2-Luc) further demonstrate decreases in activity by (i) mutating the distal MYC binding site, (ii) silencing MUC1-C, and (iii) downregulating MYC (new Figs. 3e and 3f). These findings provide evidence that MUC1-C/MYC complexes occupy and activate the BRN2 promoter and thereby induce BRN2 transcription (Results, pp. 7-8).

5. Importantly, whether or not MUC1-C induces NEPC through BRN2 is unclear. Both loss of function and gain of function as well as rescue experiments should be performed.

As suggested, we show that MUC1-C drives the BRN2 pathway in both gain-of-function and loss-of-function studies (revised Figs. 5c-5e). Previous findings from BRN2 knockdown and rescue experiments demonstrated BRN2 induction of SOX2 and NE marker expression. In concert with those results, we silenced BRN2 and found downregulation of the NE-associated ASCL1 marker (new Fig. 5h). By contrast, silencing BRN2 had no apparent effect on MYCN and EZH2 (new Fig. 5h), which are driven by MUC1-C signaling. In addition, silencing BRN2 was associated with suppression of SOX2, but not MYC, KLF4 or OCT4 expression (new Fig. 5h), indicating that, like MYCN and EZH2, MUC1-C also drives these pluripotency factors by BRN2-independent mechanisms (Results, p. 9).

6. NEPC often links to tumor aggressiveness such as cell metastasis in patients. If MUC1-C affects the transition from CRPC to NEPC, the authors should focus on invasive and metastasis assay in cells and mice, but not only cell growth assay. Since the authors already performed the xenograft assay, IHC of NE markers should be examined in xenografts.

As requested, we report that silencing MUC1-C in LNCaP-AI (new Fig. 1f) and DU-145 (new Fig. 4g) cells significantly decreases their invasion as assessed in transwell assays. Regarding studies in our xenograft models, which have not been associated with detectable metastases, we have used immunoblot analysis of tumor lysates to show that targeting MUC1-C *in vivo* decreases expression of the NE markers and the Yamanaka pluripotency reprogramming factors (revised Figs. 6b, 6g and 7d).

7. In Figure 7, the relationship between MUC1-C and BRN2 in hormone naïve PCa, CRPC and NEPC tissues should be examined systematically.

In response, further analysis showed that MUC1 high CRPC tumors associate with decreased AR, KLK3, TMPRSS2, HERC3 and NKX3-1 expression levels (new Fig. 8c). These studies were extended by

analysis of a comprehensive genomic and transcriptomic dataset generated from 429 patients with advanced prostate cancer. The results showed that MUC1 expression significantly associates with decreases in PSA/KLK3 (new Fig. 8d) and increases in BRN2 (new Fig. 8e), providing support for the central premise that MUC1-C is upregulated in PC progression in association with downregulation of AR signaling and induction of the BRN2 pathway. BRN2 overexpression in CRPC cells drives SOX2, induces NE markers and enriches for an NEPC gene signature. Here, we found that MUC1 is also significantly associated with SOX2 expression (new Fig. 8f) and the NEPC score (new Fig. 8g), an RNA-based NE expression signature, supporting a MUC1-C→BRN2→SOX2 pathway in driving NE differentiation in advanced prostate cancers (Results, p. 11).

Reviewer #2

1. Is MUC1 a true oncogene- that is, is its over expression in tumors the driving force in transforming a normal epithelial cell to a cancerous one, as very elegantly shown by Kufe and coworkers in this paper and in many others from his group?

Ectopic expression of MUC1 in 3Y1 fibroblasts was found to be sufficient for induction of anchorage-independent growth and tumorigenicity, consistent with an oncogenic function. Subsequent work showed that the MUC1-C subunit is an oncoprotein, which confers multiple hallmarks of the cancer cell, including EMT, the CSC state, epigenetic reprogramming, drug resistance and immune evasion. The present study extends this oncogenic role by further demonstrating that MUC1-C drives lineage plasticity of PC cells in association with expression of the Yamanaka reprogramming factors (OSKM; OCT4, SOX2, KLF4 and MYC), which induce pluripotency and dedifferentiation of somatic cells. These points are highlighted in the Discussion (pp. 14-15).

2. The equation presented in this and many others from the Kufe group is that high MUC1 expression is the direct cause of the malignant transformation, and that interference with the MUC1 signalling pathway will inhibit growth of the malignant cell. However, there are numerous studies showing very high expression of the MUC1 protein in perfectly normal human cells. Two examples that come to mind are the epithelial cells lining the distal tubules of the kidney, as well as breast epithelial cells in the completely normal lactating human breast. In both of these instances, the completely normal epithelial cells express exceptionally high levels of MUC1 protein. And there are many other examples. Should the high expression of the MUC1-C protein lead to malignant transformation, the thesis presented in this paper, surely one would expect that these breast and kidney cells be

transformed- they are not. This goes strongly against the simple equation of a cause and effect relationship, and suggests that high MUC1 expression is not simply transforming the cell into a malignant one. The equation must therefore be much more complex than as simplistically presented here.

In agreement, the upregulation of MUC1 expression per se is clearly not a transforming event. As noted, MUC1 is highly expressed in the lactating mammary gland, which (i) occurs in association with suppression of the EMT program to preserve epithelial integrity and differentiation and (ii) upon remodeling during involution has not been linked to breast cancer. In cancer cells, MUC1-C is upregulated by auto-inductive circuits resulting from interactions with proinflammatory TFs, such as STAT3 and NF- κ B, which are activated by stress and drive the EMT program. These findings provide support for the notion that MUC1-C contributes to cancer progression, at least in part, in association with the response to stress and inflammation, activation of an EMT program and induction of pluripotency to promote wound healing and maintain tissue homeostasis. Viewed in this way and given the association of prostatitis with EMT and prostate cancer, prolonged activation of MUC1-C in settings of chronic inflammation and, in turn, the induction of stemness and reprogramming could hold important implications for cancer progression. These points are highlighted in the Discussion (pp. 14-15).

3. Secondly, many studies have shown that interferon is a potent inducer of MUC1 expression, and very elegant studies have shown the presence of interferon responsive elements in the MUC1 promoter leading to marked increased MUC1 expression by interferons. Are oncogenes induced by interferon treatment of cells? Invariably, interferons are cell cytokines that usually inhibit cell growth and do not induce expression of oncogenes. Again, these well-established findings go against the thesis that MUC1 is an oncogene, and must be dealt with in the present paper.

In response, we treated LNCaP-AI cells with IFN- γ and found no apparent effect on MUC1-C expression or cell growth. Previous work has shown that the effects of IFN- γ treatment on MUC1 expression in PC cell lines are dependent on cell context (O'Connor JC, Prostate Cancer and Prostatic Diseases, 2005). We would therefore respectfully propose that IFN- γ studies of MUC1 and PC progression are beyond the scope of the present work and would be a focus of subsequent study.

A. LNCaP-AI**B. LNCaP-AI**
A. Lysates from LNCaP-AI cells treated with 50 ng/ml IFN- γ for the indicated times were immunoblotted with antibodies against MUC1-C and β -actin. B. LNCaP-AI cells cultured in the absence and presence of 50 ng/ml IFN- γ for 72 h were monitored for cell growth. Cell number (mean \pm SD of three replicates) was determined by trypan blue staining.

4. It is somewhat glaring in that all references supporting the thesis that MUC1 is an oncogene are those emanating from the laboratory of the author of the submitted paper. For example, references 11, 12, 13 and 14 are the sole citations supporting the statement in the Introduction "...and (ii) an oncogenic C-terminal transmembrane subunit (MUC1-C)", and again in the Results section references 30, and 31 are the sole citations for "...MUC1-C activates myc in certain cancer cells". One would feel much more comfortable with these findings were there to be completely independent confirmations reported by other research groups. The findings are sufficiently important to have warranted confirmatory experiments by other groups.

The concept that MUC1 is oncogenic has not been a central focus of the cancer field, perhaps as a result of preconceived notions that mucins are expressed to simply provide a mucous gel barrier for the protection of epithelia. Emerging evidence from us and others supports the premise that the MUC1-C subunit is an oncoprotein responsible for driving plasticity and dedifferentiation of the cancer cell. We anticipate that these findings will increase interest in studying involvement of the MUC1-C subunit in cancer.

Regarding MYC interactions, early studies by others showed that targeting full-length MUC1 suppresses MYC expression in different types of cancer cells. The mechanistic basis for those observations was elucidated by our findings that MUC1-C activates the MYC promoter, which have been confirmed by other groups. This information and the supporting references have been included in the Results (p. 7) and Discussion (p. 13).

5. Specific critiques relate to the following exceptionally important findings reported in Figure 6, that of inhibition of tumor cell growth by the peptide GO-203 (Genus Oncology-203):
- (a) there is no control experiment performed with a mutated peptide that does NOT interfere with MUC1 dimerization. This control mutated peptide should preferably be of the same size as GO-203, and be as similar as possible to

GO203 in its general makeup, by harboring only one or two mutations of critical amino acids.

- (b) there is no concentration titration experiment reported for the GO203 peptide.

In response, the GO series of MUC1-C peptide inhibitors were first evaluated in human prostate cancer models using GO-201 and CP-1, an identical control peptide with the exception that the critical CQC motif is mutated to AQA (ref. 44). The following are representative experiments showing that GO-201 is effective at different dosing schedules. In contrast, CP-1 had no apparent effect on tumor growth or histology.

DU-145

Four to six week old male Balb-c nu/nu mice were injected subcutaneously in the flank with 1×10^7 DU-145 cells. When tumors were $\sim 225 \text{ mm}^3$ (range: 200-275 mm^3) the mice were pair matched into groups of 6 and injected intraperitoneally with PBS each day (vehicle control; blue squares), 30 mg/kg GO-201 each day for 21 d (green circles) or 30 mg/kg CP-1 each day for 21 d (red triangles). Results are expressed as the mean tumor volume with a SD of <15% (left). There was no evidence of weight loss in any of the groups. Tumors harvested on day 21 from the control, GO-201 and CP-1 treatment groups were stained with H&E (right).

PC3

Male Balb-c nu/nu mice were injected subcutaneously in the flank with 1×10^7 PC3 CRPC cells. The mice were pair-matched into groups of 10 when the tumors reached $\sim 200 \text{ mm}^3$ (range: $175\text{-}250 \text{ mm}^3$). The mice were injected intraperitoneally with PBS each day for 28 d (brown squares), 30 mg/kg GO-201 each day x 28 d (blue circles), 30 mg/kg GO-201 each day for 5 d/week x 4 weeks (red circles) or 30 mg/kg CP-1 each day x 28 d (green triangles) (left). There was no weight loss in any of the groups. Tumors were harvested on day 28 and stained with H&E (right).

The GO-201 and GO-203 peptides, which both contain the CQCRRKN sequence and block MUC1-C dimerization, have identical activity *in vitro* and *in vivo*, and have been used interchangeably (ref. 45). This information and supporting references have been included in the Results section (p. 10).

Reviewers' Comments:

Reviewer #1:

Remarks to the Author:

The authors have adequately addressed all my concerns. The revised manuscript is acceptable for publication in NC.

Reviewer #2:

Remarks to the Author:

I have carefully looked through the authors responses to my critique of the paper.

The major point (bold highlighted) that needed to be addressed was in point 5 of my review, namely (quoted from my review):

"Specific critiques relate to the following exceptionally important findings reported in Figure 6, that of inhibition of tumor cell growth by the peptide GO-203 (Genus Oncology-203):

(b) there is no concentration titration experiment reported for the GO203 peptide. "

Were this point adequately addressed, that is that a titration experiment be performed with the "tumor inhibitory" GO203 peptide, wherein tumor inhibitory activity be observed at high doses and then tapering off and disappearing at lower doses, the manuscript would be acceptable for publication.

This clearly has not been done in the present revised manuscript, and the author's rebuttal of (quote)

"The following are representative experiments showing that GO-201 is effective at different dosing schedules" in no way addresses the issue of a titration experiment.

The differing dosing schedules cited by the corresponding author ("The following are representative experiments showing that GO-201 is effective at different dosing schedules") relate to either, (quote from authors rebuttal) "30 mg/kg GO-201 each day for 21 d (green circles) or 30 mg/kg CP-1 each day for 21 d (red triangles)" for the DU145 tumor, or (quote from authors rebuttal) "30 mg/kg GO-201 each day x 28 d (blue circles), 30 mg/kg GO-201 each day for 5 d/week x 4 weeks (red circles) or 30 mg/kg CP-1 each day x 28 d (green triangles) (left)" for the PC3 tumor. We see from this that the "tumor inhibitory" GO201 peptide has been administered in ALL cases at 30mg/kg for time spans ranging from every day for 20 days (5 d/week x 4 weeks) to every day for 28 days. These are NOT titration experiments and this dosage would be equivalent to administering the 24-mer GO-201 peptide to humans at a dosage of at least 2100 milligrams every day for at least 20 days for a total of 42 grams, which would be a colossal and hardly tolerable dose. Because of this, and because the authors wish to use the in-vivo mouse tumor model to show the efficacy of the GO-201(or GO-203) peptides in inhibiting tumor growth, the proposed titration experiments are absolutely critical, for indicating efficacy in translating their remarkable murine in-vivo results into a clinical setting.

It is for this reason that I find the issue of titration experiments for the tumor inhibitory activity of the GO-201/203 peptides to be of critical importance vis-à-vis accepting or rejecting the manuscript.

Without this data, the manuscript is unacceptable for publication. Indeed its publication in your prestigious Journal would grant a "stamp of approval" that these GO peptides inhibit in-vivo tumor growth via interference with MUC1-mediated signalling pathways, and thereby, in turn, grant a "stamp of approval" for the clinical use of these Genus Oncology (GO) peptides, when in fact, without the titration experiments, we are in no position to know if this is the truth.

The comments of Reviewer #2 have been addressed as follows:

1. The major point (bold highlighted) that needed to be addressed was in point 5 of my review, namely (quoted from my review): “Specific critiques relate to the following exceptionally important findings reported in Figure 6, that of inhibition of tumor cell growth by the peptide GO-203 (Genus Oncology-203): (b) there is no concentration titration experiment reported for the GO203 peptide.” Were this point adequately addressed, that is that a titration experiment be performed with the ”tumor inhibitory” GO203 peptide, wherein tumor inhibitory activity be observed at high doses and then tapering off and disappearing at lower doses, the manuscript would be acceptable for publication.

In response and in concert with the development of new agents, such as GO-201, we initially invested considerable effort in defining doses and schedules with effective therapeutic indices before reporting activity against human breast and prostate cancer models in 2009 (Raina D, Cancer Res., 69:5133-41, 2009; Joshi MD, Mol. Cancer Ther., 8:3056-65, 2009). These studies collectively defined an active GO-201 dose of 30 mg/kg/d that was well tolerated. Other groups have shown that (i) dosing GO-201 at 30 mg/kg/d is highly effective against human AsPC-1 pancreatic tumor xenografts (Banerjee S, PLoS One, e43020, 2012), and (ii) dosing GO-201 at 15 mg/kg/d results in partial inhibition of human esophageal squamous cell carcinoma (ESCC) tumor growth (Xin Z, OncoTargets and Therapy, 11:4125-36, 2018). Statements to this effect with citations of the supporting references have been included in the Results (p. 10).

These findings with GO-201 were extended in titration studies of GO-203, which showed partial tumor inhibitory activity at a dose of 20 mg/kg/d and complete tumor regressions at 30 mg/kg/d. Consistent with a dose-dependent response, GO-203 doses of 2.5, 5 and 10 mg/kg/d had little if any tumor inhibitory effect. In addition, and as found for GO-201, treatment with GO-203 at 30 mg/kg/d was well tolerated without evidence of weight loss or other overt toxicities.

Titration studies of GO-203 demonstrate dose-dependent anti-tumor activity. Female Balb/c nu/nu mice were implanted with 17- β -estradiol plugs. After 24 h, ZR-75-1 cells (1×10^7) derived from xenografts were injected sc in the flank. When tumors reached 140-170 mm³, the mice were pair matched into groups of 8 each and injected ip with PBS (brown triangles) or GO-203 at doses of 2.5 mg/kg/d (blue squares), 5 mg/kg/d (red diamonds), 10 mg/kg/d (green circles), 20 mg/kg/d (open triangles) and 30 mg/kg/d (black triangles) for 21 days. Mice were weighed twice weekly and tumor measurements were performed every two days. The results are expressed as mean tumor volumes (SEMs of <10%). There was no evidence of weight loss in the treated groups.

Dosing of GO-203 at 30 mg/kg/d has also been shown to be effective in suppressing the growth of human H1975 and A549 non-small cell lung cancer (NSCLC) xenografts (Raina D, Mol. Cancer Ther., 10:806-16, 2011). Additionally, the dose-dependent effects of GO-203 have provided experimental settings for evaluating GO-203 in combination with other anti-cancer drugs. In this way, GO-203 doses of 12 mg/kg/d partially inhibit NSCLC tumor growth and have been used to assess GO-203 combinations with targeted agents, such as afatinib and JQ1 (Kharbanda A, Clin. Cancer Res., 20:5423-34, 2014; Bouillez A, Cancer Res., 76:1538-48, 2016). GO-203 doses of 15 mg/kg/d have also been used by others for partial inhibition of ESCC tumor growth (GongSun X, J. Cellular Physiol., 234:12019-28, 2019), supporting dose-dependent effects against diverse human cancers. Statements regarding these effects of GO-203 with citations of the supporting references have been included in the Results (p. 10).

As noted in Discussion, daily administration of GO-203 has entered evaluation in early phase clinical trials and has also been formulated in nanoparticles (GO-203/NP) for more convenient dosing schedules of once or twice a week. As shown for unencapsulated GO-203, titration studies with the GO-203/NPs confirmed dose-dependent activity in preclinical models to support development of this formulation in IND enabling studies (Hasegawa M, Clin. Cancer Res., 21:2338-47, 2015).

Dose-dependent activity of GO-203 encapsulated in nanoparticles. Balb/c mice bearing subcutaneous Ehrlich tumors (~40 mm³) were treated IP with vehicle control (closed squares), 10 mg/kg (closed circles), 15 mg/kg (open circles) or 20 mg/kg (closed triangles) GO-203/NPs once a week for 3 weeks. Tumor volumes were determined on the indicated days of treatment. The results are expressed as tumor volumes (mean \pm SEM for 10 mice in each group).

This information regarding the dose-dependent effects of GO-203/NP administration has been included in the Results (p. 10) and Discussion (p. 15).

2. This dosage would be equivalent to administering the 24-mer GO-201 peptide to humans at a dosage of at least 2100 milligrams every day for at least 20 days for a total of 42 grams, which would be a colossal and hardly tolerable dose.

We would respectfully contend that the above dosing calculations for humans are incorrect. The formula for determining a mouse to human equivalent dose would be:

Mouse dose (mg/kg) x mouse factor (3) = human dose (mg/kg) x human factor (37)

Therefore, based on the GO-203 dose of 30 mg/kg in mice:

$(30 \text{ mg/kg}) \times 3 = 90/37 = \text{human dose of } 2.43 \text{ mg/kg.}$

The average human is ~70 kg; therefore $70 \times 2.43 = 170.1 \text{ mg/d}$, NOT the above projected 2100 mg/d, which is more than 10-fold higher. Indeed, a GO-203 dose of 170 mg/d in humans is compatible with that for certain approved targeted agents (i.e., sorafenib is administered at 800 mg/d).

Reviewers' Comments:

Reviewer #2:

Remarks to the Author:

Accept for publication the revised manuscript.